# Quantifying and Alleviating Co-Adaptation in Sparse-View 3D Gaussian Splatting

**Kangjie Chen**[1][*]   **Yingji Zhong**[2]    **Zhihao Li**[3]    **Jiaqi Lin**[1]
**Youyu Chen**[4]    **Minghan Qin**[1]    **Haoqian Wang**[1][†]

[1]Tsinghua University    [2]HKUST    [3]Huawei Noah's Ark Lab
[4]Harbin Institute of Technology

## Abstract

3D Gaussian Splatting (3DGS) has demonstrated impressive performance in novel view synthesis under dense-view settings. However, in sparse-view scenarios, despite the realistic renderings in training views, 3DGS occasionally manifests appearance artifacts in novel views. This paper investigates the appearance artifacts in sparse-view 3DGS and uncovers a core limitation of current approaches: the optimized Gaussians are overly-entangled with one another to aggressively fit the training views, which leads to a neglect of the real appearance distribution of the underlying scene and results in appearance artifacts in novel views. The analysis is based on a proposed metric, termed Co-Adaptation Score (CA), which quantifies the entanglement among Gaussians, i.e., co-adaptation, by computing the pixel-wise variance across multiple renderings of the same viewpoint, with different random subsets of Gaussians. The analysis reveals that the degree of co-adaptation is naturally alleviated as the number of training views increases. Based on the analysis, we propose two lightweight strategies to explicitly mitigate the co-adaptation in sparse-view 3DGS: (1) random gaussian dropout; (2) multiplicative noise injection to the opacity. Both strategies are designed to be plug-and-play, and their effectiveness is validated across various methods and benchmarks. We hope that our insights into the co-adaptation effect will inspire the community to achieve a more comprehensive understanding of sparse-view 3DGS.

## 1  Introduction

Recent advances in 3D Gaussian Splatting (3DGS) [1] have demonstrated remarkable capabilities for photorealistic novel view synthesis in dense-view settings. However, under sparse-view supervision, 3DGS often suffers significant performance degradation in novel view rendering, manifesting as artifacts caused by incorrect geometry or appearance distribution due to limited supervision from training views. Existing works mainly improve sparse-view 3DGS through geometry regularization, such as monocular depth constraints [2, 3, 4, 5, 6, 7, 8, 9], matching-based point initialization [10], and dense point initialization [11, 12]. Although these methods achieve improvements by enhancing geometric accuracy, few have investigated appearance artifacts in sparse-view 3DGS. These artifacts, often overlooked, are common in novel view rendering and typically manifest as occasional color outliers—colors not belonging to the scene—as illustrated in Figure 1. In this paper, we focus on analyzing and mitigating appearance artifacts in the novel view rendering of sparse-view 3DGS.

We begin our analysis by revisiting the rendering characteristics of 3DGS. In the splatting process, each pixel receives contributions from multiple Gaussians projected onto its corresponding tile on the

---

[*]This work was done while he was an intern at Huawei.

[†]Corresponding author. E-mail: `wanghaoqian@tsinghua.edu.cn`

39th Conference on Neural Information Processing Systems (NeurIPS 2025).

**Why do we see colors that don't belong to the scene?**

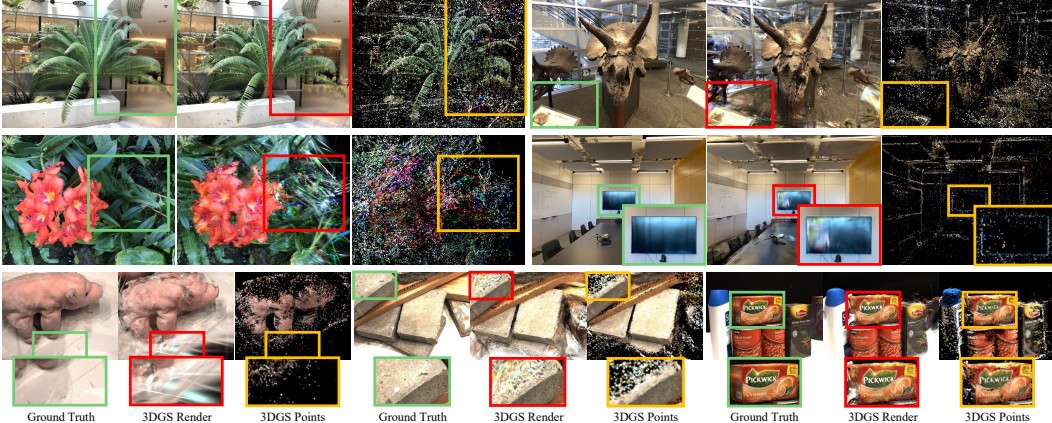

Figure 1: **Appearance artifacts in sparse-view 3DGS**. In novel view rendering, the optimized 3DGS occasionally produces colors that do not match the scene's true appearance distribution.

image plane [1]. This multi-Gaussian composition is a fundamental characteristic of 3DGS, enabling effective appearance modeling under dense-view settings. However, it also introduces potential entanglement between Gaussians, especially when they are jointly optimized to fit overlapping pixel regions in the training views. In sparse-view scenarios, such entanglement becomes problematic—multiple Gaussians with divergent colors may aggressively collaborate to fit limited training pixels, ignoring the true underlying appearance distribution. Here, we refer to this entanglement as co-adaptation in 3DGS, and hypothesize that appearance artifacts in sparse-view novel view synthesis are closely related to excessive co-adaptation.

To validate our hypothesis, we propose a metric to quantify the level of co-adaptation across Gaussians, namely the Co-Adaptation score (CA). Specifically, we randomly select subsets of Gaussians using dropout and render the same target view multiple times. The CA score is computed as the average per-pixel variance across these renderings. In the appendix, we further provide a theoretical derivation showing that the CA score directly reflects the coupling between the color and opacity attributes of Gaussians. Empirically, we observe that the CA score decreases as the number of training views increases. This suggests that the degree of co-adaptation in 3DGS tends to be naturally suppressed as the training data becomes denser. This observation inspires us to further investigate whether explicitly reducing co-adaptation in sparse-view 3DGS can improve novel view rendering quality.

To mitigate co-adaptation in sparse-view 3DGS, we propose two simple yet effective strategies. (1) We apply dropout during training by sampling a subset of Gaussians, disrupting excessive entanglement among Gaussians. (2) We inject multiplicative noise into the opacity parameters throughout training, perturbing each Gaussian's contribution to the rendered pixel. Unlike perturbing position or color, which can harm convergence, or perturbing scale, which tends to cause blurring, opacity noise provides softer, more targeted regularization by destabilizing the reliance structure among co-adapted splats. Both strategies are plug-and-play, and their effectiveness is validated across various methods and benchmarks in the experiment section.

To summarize, our contributions are as follows: (1) We analyze the source of appearance artifacts and identify the entanglement among Gaussians as the performance bottleneck for sparse-view 3DGS. (2) We propose a metric of Co-Adaptation Score (CA), to quantitatively measure the entanglement among Gaussians in sparse-view 3DGS. (3) Based on the analysis and the proposed metric, we further propose two plug-and-play training strategies that can suppress co-adaptation among gaussians, i.e., random gaussian dropout and multiplicative noise injection, whose effectiveness is validated across various methods and benchmarks.

## 2 Related Work

**3D Gaussian Splatting under Sparse Views.** Recent studies have explored a variety of strategies to improve 3D Gaussian Splatting (3DGS) [1] under sparse-view supervision. Some approaches [2, 3, 4,

5, 6, 7, 8, 9] incorporate external geometry priors, typically from monocular depth estimators [13, 14], to enforce multi-view depth consistency and enhance geometric fidelity. Other works [15, 16, 17, 18, 19, 20, 21] leverage generative priors from diffusion models to synthesize plausible unseen views or guide view-dependent effects. In addition, other methods [10, 22, 23] introduce self-supervised or physically motivated regularization strategies, such as opacity decay, binocular-guided photometric consistency, and dual-model co-regularization, to suppress floating artifacts and improve geometric consistency. Additionally, a separate line of work [4, 6, 24, 25, 26, 27, 28, 29, 30] departs from per-scene optimization by training neural predictors across large-scale data to directly infer 3DGS representations from sparse inputs. Recent works [31, 32] have also explored dropout-based strategies for improving sparse-view 3DGS performance. While both of them report clear improvements, they attribute the effectiveness of dropout to empirical factors—such as reducing overfitting through fewer active splats [31], or enhancing gradient flow to distant Gaussians [32]. In contrast, our work identifies and formalizes co-adaptation suppression as the key underlying mechanism, offering a more principled explanation for why dropout benefits sparse-view generalization in 3DGS.

**Co-Adaptation in Neural Networks.** Co-adaptation describes the tendency of neurons in neural networks to become overly dependent during training, leading to overfitting and poor generalization. This issue was first highlighted by Hinton et al. [33], who proposed Dropout to mitigate such dependency by randomly deactivating neurons. Subsequent stochastic regularization methods [34, 35, 36, 37, 38, 39, 40] further promote feature diversity through randomization, consistency enforcement, or latent-space mixing. Beyond empirical methods, theoretical studies [41, 42, 43] have analyzed co-adaptation's negative impact on generalization, linking it to sharp loss minima [41], low sensitivity to perturbations [42], and poor uncertainty estimation [43]. While these insights have been validated in classification and regression tasks, their implications for structured prediction like 3D reconstruction remain underexplored. Our work extends this investigation to 3D Gaussian Splatting, revealing similar interdependencies among spatial Gaussians and showing that dropout and opacity noise effectively mitigate co-adaptation under sparse-view supervision.

## 3 Co-Adaptation in Gaussian Splatting

In this chapter, we investigate co-adaptation in 3D Gaussian Splatting (3DGS) [1], which significantly affects generalization under sparse-view settings. First, we explain co-adaptation in 3DGS and analyze why excessive co-adaptation leads to artifacts in novel view synthesis. Then, we propose a metric to quantify co-adaptation via variance across dropout-rendered outputs. Based on this metric, we make empirical observations about co-adaptation in 3DGS. Finally, we introduce two simple strategies to explicitly mitigate co-adaptation: dropout regularization and opacity noise injection.

### 3.1 Characterizing Co-Adaptation

Unlike explicit spatial representations [44, 45, 46] that define fixed scene geometry, 3DGS represents scenes as a set of view-dependent Gaussians without explicit surface definitions. This allows 3DGS to model flexible, appearance-driven representations that adapt to different viewpoints. Specifically, 3DGS renders images by projecting 3D Gaussians onto the image plane and blending their contributions using differentiable alpha compositing. The color of each pixel $u$ is computed as:

$$C(u) = \sum_{i \in \mathcal{N}(u)} c_i \, \alpha_i \prod_{j=1}^{i-1} (1 - \alpha_j), \tag{1}$$

where $\mathcal{N}(u)$ denotes the set of Gaussians projected to pixel $u$, sorted by depth. Each $\alpha_i$ is the projected opacity of the $i$-th Gaussian, serving as its blending weight. As shown in Equation 1, the perceived color $C(u)$ depends on the specific combination of Gaussians $\mathcal{N}(u)$ along the rendering ray. While this cooperative blending mechanism makes 3DGS highly effective in fitting the appearance of training views, its optimization objective focuses solely on minimizing the discrepancy between the rendered images and the ground-truth views. Specifically, 3DGS optimizes the Gaussian set $\mathcal{G}$ to minimize the reconstruction loss over training views $\mathcal{V}_{\text{train}}$:

$$\mathcal{G}^* = \arg \min_{\mathcal{G}} \sum_{v \in \mathcal{V}_{\text{train}}} \mathcal{L}(R(\mathcal{G}, v), I_v), \tag{2}$$

where $R(\mathcal{G}, v)$ represents the rendered image from view $v$ using $\mathcal{G}$, and $I_v$ is the ground-truth image. The loss $\mathcal{L}$ supervises only the rendered outputs, without imposing any explicit internal constraints

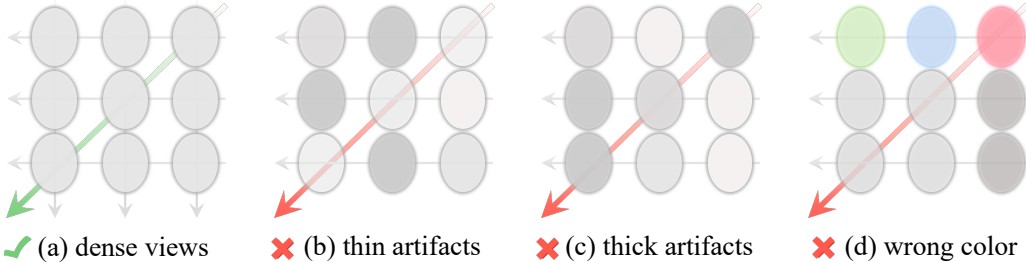

| ✔ (a) dense views | ✘ (b) thin artifacts | ✘ (c) thick artifacts | ✘ (d) wrong color |

Figure 2: **Visualization of 3DGS behaviors under different levels of co-adaptation.** Thin gray arrows indicate training views, bold arrows indicate a novel view. Green arrow denotes correct color prediction, while red indicates color errors. (a) Simulates a 3DGS model trained with dense views, where Gaussian ellipsoids contribute evenly to pixel color across views, resulting in accurate rendering from the novel view. (b)(c)(d) Simulate various cases of 3DGS trained under sparse-view settings. (b) and (c) show that co-adaptation in the training views — where Gaussians contribute unequally to pixel colors — results in thin and thick artifacts under novel views. (d) shows a highly co-adapted case where multiple Gaussians with distinct colors collectively overfit a single grayscale pixel in the training view, resulting in severe wrong color artifacts under the novel view.

on the Gaussian parameters such as position, shape, color, or opacity. As a result, multiple Gaussians can easily form tightly coupled combinations to fit each pixel in the training views. This dependency becomes especially problematic under sparse-view supervision (see Figure 1), where the model tends to overfit fragile Gaussian configurations that fail to generalize to novel views.

Figure 2 further illustrates this phenomenon, showing how diverse Gaussians with different colors collaborate to reproduce grayscale pixels in training views, yet produce inconsistent colors in unseen views. We refer to this phenomenon as *co-adaptation*. To systematically understand and mitigate this effect, it is essential to establish a quantitative metric that captures the severity of co-adaptation.

## 3.2 Quantifying Co-Adaptation

To quantitatively analyze co-adaptation in 3D Gaussian Splatting, we define a *co-adaptation score* for each target viewpoint. The key idea is that if a set of Gaussians are overly dependent on each other, then randomly removing part of them during rendering will lead to unstable outputs. Specifically, we randomly drop 50% of the Gaussians and render the target view using only the remaining ones. We repeat this process multiple times and measure the variance across the rendered results.

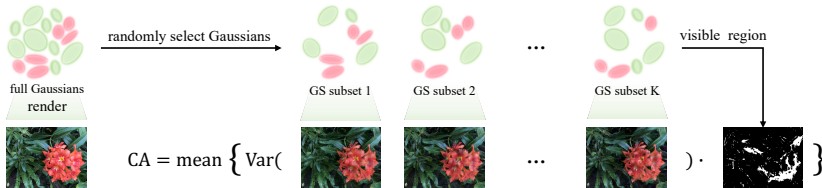

Figure 3: **Illustration of Co-Adaptation Score (CA) Computation.** Higher CA scores indicate more inconsistent renderings, suggesting stronger co-adaptation effects. Lower CA scores reflect more stable and generalizable representations.

To focus on regions where Gaussians contribute meaningfully, we define the *visible region* as the set of pixels whose accumulated alpha-blending weight exceeds a threshold of 0.8 in each rendering. The final co-adaptation score is computed as the average pixel-wise variance within the intersection of these visible regions across all $K$ random dropout renderings. Formally, the score for a given viewpoint $v$ is defined as:

$$\mathrm{CA}(v) = \frac{1}{|\Omega_v|} \sum_{u \in \Omega_v} \mathrm{Var}\left(I_u^{(1)}, \ldots, I_u^{(K)}\right), \tag{3}$$

where $I_u^{(k)}$ denotes the color at pixel $u$ in the $k$-th dropout-rendered image. To ensure that comparisons are made over consistently visible regions, we define $\Omega_v = \bigcap_{k=1}^{K} \left\{ u \mid \alpha_u^{(k)} > 0.8 \right\}$ denotes the set

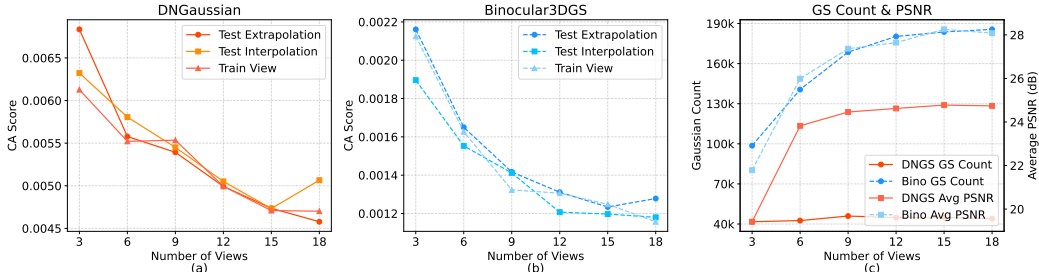

Figure 4: **Comparison of co-adaptation strength (CA), GS count and reconstruction quality under varying numbers of training views.** (a-b) CA measured as pixel-wise variance for DNGaussian and Binocular3DGS on three target view types: extrapolation (far), interpolation (near), and training. (c) Gaussian count (left axis, in thousands) and average PSNR (right axis, in dB) for both methods. All plots are shown as functions of input view counts, based on the LLFF "flower" scene.

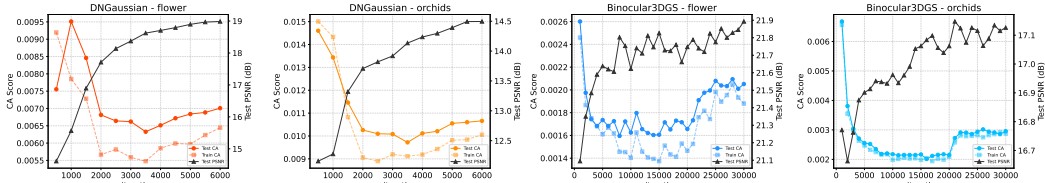

Figure 5: **Training dynamics of co-adaptation strength (CA) and reconstruction quality (PSNR) across different LLFF scenes.** CA score (left axis) and PSNR (right axis) over training iterations for DNGaussian and Binocular3DGS on the "flower" and "orchids" scenes of the LLFF dataset.

of commonly visible pixels, and $\alpha_u^{(k)} = \sum_{i \in \mathcal{N}_u^{(k)}} \alpha_i \prod_{j=1}^{i-1}(1 - \alpha_j)$ is the accumulated alpha via front-to-back compositing. $\mathcal{N}_u^{(k)}$ represents the ordered list of Gaussians contributing to pixel $u$ in the $k$-th rendering, sorted by depth and blended using front-to-back alpha compositing. A higher $\mathrm{CA}(v)$ score reflects stronger co-adaptation, as the rendered appearance becomes more sensitive to which Gaussians are selected. We further provide a theoretical derivation in the appendix, showing that $\mathrm{CA}(v)$ directly reflects the coupling between the color and opacity attributes of Gaussians.

**Empirical observations on Co-Adaptation score (CA) in sparse-view 3DGS.** We summarize three empirical phenomena observed during sparse-view 3DGS training:

1) **Increased training views reduce co-adaptation.** Figure 4 shows that co-adaptation scores drop consistently as the number of training views increases, for both DNGaussian [2] and Binocular3DGS [10]. While DNGaussian uses random initialization with a relatively stable Gaussian count in LLFF scenes [47], Binocular3DGS leverages a pre-trained keypoint matcher [48] to generate denser initializations with more views. Despite these differences, both methods demonstrate that increasing view counts weakens co-adaptation and improves generalization.

2) **Co-adaptation temporarily weakens during early training.** Figure 5 shows the evolution of co-adaptation scores during training. We analyze both DNGaussian and Binocular3DGS on two LLFF scenes, tracking CA scores on training and novel test views. In the early stages—typically the first few thousand iterations—CA scores drop sharply as the model rapidly fits visible content. Afterwards, scores stabilize and oscillate within a fixed range. Notably, Binocular3DGS shows a secondary increase in co-adaptation after 20k iterations, coinciding with the introduction of warp-based supervision on novel views. This supervision likely introduces geometric mismatches, reinforcing undesired dependencies among Gaussians and causing the observed rise in CA.

3) **Co-adaptation is lower at input views than novel views.** Across nearly all scenes and training iterations (see Figure 5), co-adaptation scores at input (training) views remain consistently lower than those at novel (test) views. This indicates that co-adaptation is more easily suppressed under familiar viewpoints, while novel views tend to retain stronger entanglements among Gaussians.

Inspired by these empirical findings, we investigate whether suppressing co-adaptation in 3DGS can enhance rendering quality for novel views.

### 3.3 Strategies for Alleviating Co-Adaptation

We explore two regularization strategies to mitigate excessive co-adaptation in 3D Gaussian Splatting. First, we adapt dropout [33, 49] to randomly mask subsets of Gaussians during rendering, reducing over-reliance on fixed combinations. Second, inspired by noise-based regularization [50], we inject Gaussian noise into opacity parameters, encouraging more flexible Gaussian contributions. Both techniques aim to enhance generalization by loosening overly rigid dependencies between Gaussians.

**Dropout Regularization.** Dropout is a classical regularization technique that mitigates neural network overfitting by randomly disabling a subset of model components during training. Inspired by this idea, we apply a random dropout strategy to the Gaussian set during 3DGS training. At each iteration, every Gaussian has a probability $p$ of being temporarily dropped:

$$\mathcal{G}_{\text{train}} = \{g \in \mathcal{G} \mid z_g = 1\}, \quad z_g \sim \text{Bernoulli}(1 - p) \tag{4}$$

where $\mathcal{G}$ denotes the full set of Gaussians, and $z_g$ is a binary random variable indicating whether Gaussian $g$ is retained ($z_g = 1$) or dropped ($z_g = 0$) during training. Each $z_g$ is independently sampled from a Bernoulli distribution with retention probability $1 - p$. The rendered image using the surviving subset $\mathcal{G}_{\text{train}}$ is then supervised against the ground truth of the corresponding training view. At test time, all Gaussians are used, but we scale their opacities to match training-time expectations:

$$\alpha_g^{\text{test}} = (1 - p) \cdot \alpha_g^{\text{train}}, \quad \forall g \in \mathcal{G}. \tag{5}$$

**Opacity Noise Injection.** Besides dropout, we explore injecting stochastic noise into Gaussian opacity to reduce co-adaptation. Unlike dropout, which removes entire Gaussians, opacity noise slightly perturbs each Gaussian's contribution, improving robustness to varying blending. We also tested adding noise to different parameters: 3D position noise caused instability and blur; SH coefficient noise had little effect, as SH only affects color, not visibility or Gaussian count per pixel; scale noise also reduced co-adaptation but introduced blur with limited quality improvement.

Our final formulation focuses on opacity:

$$\text{opacity} \leftarrow \text{opacity} \cdot (1 + \epsilon), \quad \epsilon \sim \mathcal{N}(0, \sigma^2). \tag{6}$$

**Discussion. (1) Dropout mechanism** forces each ray to remain accurately supervised even when part of its contributing Gaussians are randomly omitted. As a result, the model learns to avoid over-reliance on fixed Gaussian configurations, as nearby Gaussians along the ray tend to acquire similar color and opacity characteristics, making them mutually substitutable. Therefore, dropout effectively alleviates excessive co-adaptation in Gaussian Splatting. Furthermore, since some Gaussians may be randomly dropped during training, the remaining ones tend to grow larger to maintain consistent surface coverage. This encourages the model to represent the scene using larger Gaussians, which helps reduce geometric inconsistencies and surface gaps, especially in sparse-view settings. By disrupting such entanglements, dropout enhances both appearance and structural generalization. **(2) Opacity noise** encourages nearby Gaussians along a ray to exhibit consistent color and transparency characteristics, making them less dependent on specific groupings. By weakening such over-specialized configurations, this strategy improves generalization under sparse-view conditions.

## 4 Experiments

### 4.1 Experiments Setup

**Datasets.** We conduct experiments on three datasets: LLFF [47], DTU [51], and Blender dataset [52]. Following prior works [10, 53, 54, 55], we use 3 training views for LLFF and DTU, and 8 views for Blender. Test views follow prior settings [10, 53, 54, 55]. Input images are downsampled by a factor of 8 for LLFF, 4 for DTU, and 2 for Blender, relative to their original resolutions.

**Baselines.** We conduct experiments on 3D Gaussian Splatting (3DGS) [1], DNGaussian [2], FSGS [3], CoR-GS [22], and Binocular3DGS [10]. Our two proposed strategies, dropout regularization and opacity noise injection, are applied to each method to evaluate their effectiveness. Ablation studies are primarily conducted on Binocular3DGS. We follow the official implementations for all baselines. On LLFF, our reproduced results for DNGaussian and CoR-GS slightly differ from the reported values. For Blender, DNGaussian uses different training settings for different scenes, while we adopt a unified setup for fair comparison, which leads to lower results. We also re-run Binocular3DGS with

Table 1: Quantitative Comparison on LLFF and DTU Datasets. We evaluate five sparse-view 3DGS-based methods with and without our proposed co-adaptation suppression strategies, *dropout regularization* and *opacity noise injection*. We report PSNR, SSIM, LPIPS, and Co-Adaptation scores (CA) on both training and novel views to assess reconstruction quality and co-adaptation reduction.

| Method | Setting | LLFF [47] | | | | | DTU [51] | | | | |
| | | PSNR ↑ | SSIM ↑ | LPIPS ↓ | Train CA ↓ | Test CA ↓ | PSNR ↑ | SSIM ↑ | LPIPS ↓ | Train CA ↓ | Test CA ↓ |
|---|---|---|---|---|---|---|---|---|---|---|---|
| 3DGS [1] | baseline | 19.36 | 0.651 | 0.232 | 0.007543 | 0.008206 | 17.30 | 0.824 | 0.152 | 0.002096 | 0.002869 |
| | w/ dropout | **20.20** | **0.691** | **0.211** | 0.001752 | 0.002340 | **17.75** | **0.850** | **0.135** | **0.000757** | **0.002263** |
| | w/ opacity noise | 19.91 | 0.676 | 0.223 | **0.001531** | **0.002300** | 17.27 | 0.839 | 0.140 | 0.001203 | 0.002390 |
| DNGaussian [2] | baseline | 18.93 | 0.599 | 0.295 | 0.007234 | 0.007645 | 18.91 | 0.790 | 0.176 | 0.005113 | 0.005744 |
| | w/ dropout | **19.43** | **0.623** | 0.302 | **0.003242** | **0.003821** | **19.86** | **0.828** | **0.149** | **0.001201** | **0.001917** |
| | w/ opacity noise | 19.15 | 0.608 | **0.294** | 0.004507 | 0.005071 | 19.52 | 0.813 | 0.153 | 0.001901 | 0.002641 |
| FSGS [3] | baseline | 20.43 | 0.682 | 0.248 | 0.004580 | 0.004758 | 17.34 | 0.818 | 0.169 | 0.002078 | 0.003313 |
| | w/ dropout | **20.82** | **0.716** | **0.200** | 0.001930 | 0.002205 | **17.86** | **0.838** | **0.153** | **0.001057** | 0.002376 |
| | w/ opacity noise | 20.59 | 0.706 | 0.210 | **0.001666** | **0.002020** | 17.81 | 0.834 | 0.156 | 0.001096 | **0.002245** |
| CoR-GS [22] | baseline | 20.17 | 0.703 | **0.202** | 0.005025 | 0.005159 | 19.21 | 0.853 | 0.119 | 0.001090 | 0.001199 |
| | w/ dropout | **20.64** | **0.712** | 0.217 | **0.001442** | **0.001617** | **19.94** | **0.868** | 0.118 | **0.000378** | **0.000611** |
| | w/ opacity noise | 20.28 | 0.705 | **0.202** | 0.003731 | 0.003867 | 19.54 | 0.862 | **0.115** | 0.000558 | 0.000814 |
| Binocular3DGS [10] | baseline | 21.44 | 0.751 | 0.168 | 0.001845 | 0.001951 | 20.71 | 0.862 | 0.111 | 0.001399 | 0.001579 |
| | w/ dropout | **22.12** | 0.777 | **0.154** | 0.000875 | 0.000978 | 21.03 | **0.875** | 0.108 | 0.000752 | 0.001146 |
| | w/ opacity noise | **22.12** | 0.780 | 0.155 | **0.000660** | **0.000762** | 20.92 | 0.866 | **0.107** | 0.001346 | 0.001548 |
| | w/ both | 22.11 | **0.781** | 0.157 | 0.000673 | **0.000762** | **21.05** | **0.875** | 0.109 | **0.000736** | **0.001143** |

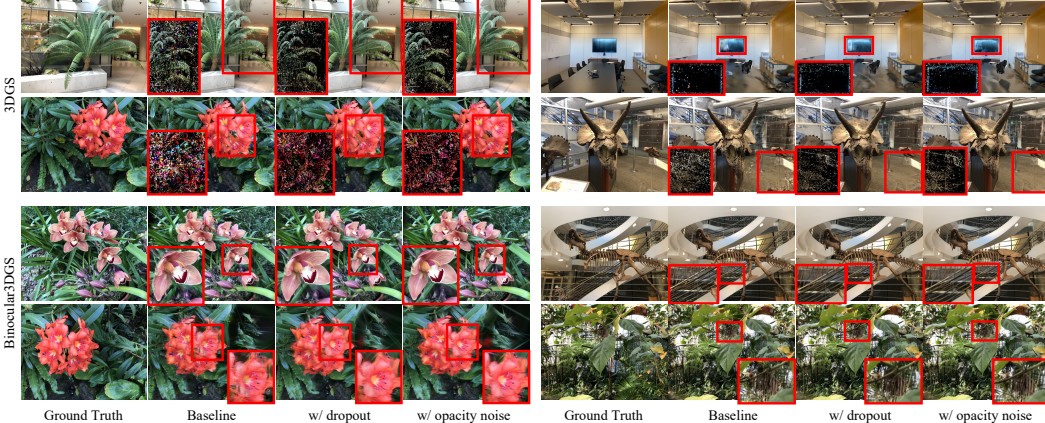

Figure 6: Visual comparison on the LLFF dataset based on 3DGS and Binocular3DGS. Suppressing co-adaptation reduces color noise and improves scene geometry and detail quality.

white backgrounds to align with other methods. Baseline results in Table 2 are reported as original / reproduced, showing the original paper values and our re-implementation under the unified setup.

**Metrics.** To evaluate the rendering quality under novel views, we compute standard image quality metrics including average PSNR (peak signal-to-noise ratio), SSIM [56], and LPIPS [57] on each dataset. To assess the impact of different strategies on co-adaptation, we additionally report Co-Adaptation Score (CA) (Section 3.2) computed on both training views and novel test views.

**Implementation Details.** To ensure consistent scaling when comparing co-adaptation scores, we adjust the drop ratio during CA computation to account for training-time dropout. Specifically, in the absence of dropout in training, CA is measured by randomly discarding 50% of the contributing Gaussians. When dropout with rate $p$ (e.g., $p = 0.2$) is used in training, we instead discard $1 - \frac{1-p}{2}$ (i.e., 60%) of the Gaussians during CA computation in test time.

## 4.2 Comparative Analysis of Co-Adaptation Suppression

**(1) Analysis of Quantitative Results on Co-Adaptation and Rendering Quality.** We evaluate the impact of our proposed dropout and opacity noise strategies across five 3DGS-based methods in sparse-view settings. As shown in Table 1, both strategies effectively reduce the Co-Adaptation Scores (CA) on both training and novel views across LLFF and DTU datasets, confirming their ability to mitigate over co-adaptation. We further observe that dropout generally outperforms opacity noise in improving rendering quality, consistently achieving higher PSNR and lower LPIPS in almost all tested settings. This suggests that dropout not only reduces co-adaptation but also better preserves novel view fidelity than opacity noise. However, applying both strategies together does not lead

Table 2: Quantitative comparison on Blender dataset for 8 input views. We evaluate three sparse-view 3DGS-based methods with and without our proposed co-adaptation suppression strategies.

| Method | Setting | PSNR ↑ | SSIM ↑ | LPIPS ↓ | Train CA ↓ | Test CA ↓ |
|--------|---------|--------|--------|---------|------------|-----------|
| 3DGS [1] | baseline | 23.54 | 0.881 | 0.095 | 0.004247 | 0.003937 |
|  | w/ dropout | **24.14** | **0.890** | **0.090** | **0.001805** | **0.002015** |
|  | w/ opacity noise | 23.65 | 0.884 | 0.094 | 0.003167 | 0.003136 |
| DNGaussian [2] | baseline | 24.31 / 23.41 | 0.886 / 0.879 | 0.088 / 0.097 | 0.004352 | 0.004947 |
|  | w/ dropout | **23.98** | **0.890** | **0.089** | **0.001973** | **0.002585** |
|  | w/ opacity noise | 23.43 | 0.878 | 0.097 | 0.004282 | 0.004922 |
| Binocular3DGS [10] | baseline | 24.71 / 23.81 | 0.872 / 0.889 | 0.101 / 0.093 | 0.001776 | 0.001981 |
|  | w/ dropout | **24.24** | **0.894** | **0.090** | 0.001383 | 0.001539 |
|  | w/ opacity noise | **24.24** | **0.894** | 0.092 | **0.001038** | **0.001179** |

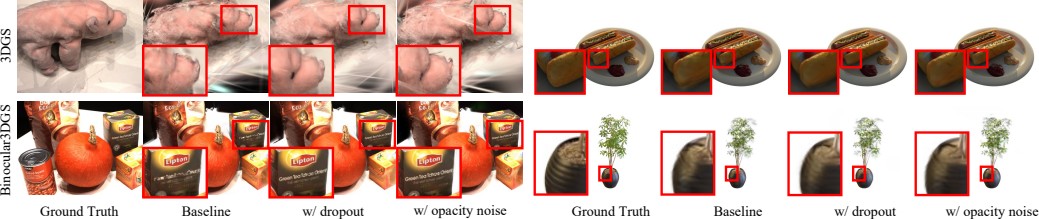

Figure 7: Visual comparison on DTU and Blender datasets based on 3DGS and Binocular3DGS. Suppressing co-adaptation leads to clearer appearance representation in novel view rendering.

to further improvements beyond using either strategy alone, indicating they address the same issue without additive benefits. Table 2 extends the comparison to the Blender dataset, where experiments are conducted on 3DGS, DNGaussian, and Binocular3DGS. On Blender, both strategies reduce CA, but the 3DGS baseline shows higher training-view CA than novel-view CA, likely due to the ring-shaped object-based scene structure and denser 8-view coverage. Random initialization and lack of geometric constraints in 3DGS may further blur the distinction between training and novel views.

**(2) Visual Improvements on Gaussian Point Clouds and Rendering Quality.** We further provide visual comparisons on LLFF, DTU, and Blender datasets. Specifically, Figure 6 presents results on the LLFF dataset, while Figure 7 shows side-by-side comparisons on DTU and Blender datasets. These visualizations highlight improvements by our strategies on LLFF, DTU, and Blender, showing reduced artifacts and clearer structures for both 3DGS and Binocular3DGS. As shown in Figure 6, both strategies effectively suppress the colorful speckle artifacts commonly observed in baseline 3DGS renderings, while also improving the spatial coherence of the reconstructed scenes. Compared to baseline 3DGS, Binocular3DGS starts with a denser, keypoint-guided initialization, which already reduces such artifacts. Nevertheless, applying our strategies further improves the rendered results, producing more complete object appearances—for example, clearer flower petals in the flower and orchids scenes, and sharper boundaries in the trex scene. We emphasize that while Binocular3DGS produces fewer visible artifacts, co-adaptation remains an underlying challenge affecting generalization to novel views. Colorful speckles are merely one extreme symptom of this issue. Our strategies consistently reduce these artifacts and improve rendering quality across both baselines. Additionally, Figure 7 demonstrates that these benefits generalize to DTU and Blender datasets. On DTU and Blender, dropout and opacity noise lead to clearer and more stable structures.

## 4.3 Ablation Studies of Regularization Strategies

**(1) Training Dynamics Comparison.** Figure 8 compares the training dynamics of Binocular3DGS baseline with our Dropout and Opacity Noise Injection strategies on flower and orchids scenes. Both strategies reduce the CA scores of training and testing views, making the dynamics smoother than the baseline. Although the baseline shows a sharp CA increase around 20k iterations due to adding a warp-based photometric loss, the two strategies help suppress this spike, especially on the flower scene where no significant rise is observed. On the orchids scene,

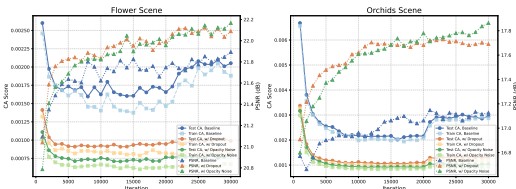

Figure 8: Training dynamics (CA and PSNR) comparison of Baseline and w/ two training strategies.

Table 3: Ablation study on the dropout probability $p$ in Binocular3DGS on the LLFF dataset.

| Dropout $p$ | PSNR ↑ | SSIM ↑ | LPIPS ↓ | Train CA ↓ | Test CA ↓ |
|---|---|---|---|---|---|
| 0.0 (Baseline) | 21.440 | 0.751 | 0.168 | 0.001845 | 0.001951 |
| 0.1 | 21.901 | 0.768 | 0.157 | 0.000995 | 0.001066 |
| 0.2 | **22.123** | **0.777** | **0.154** | 0.000875 | 0.000978 |
| 0.3 | 22.037 | **0.777** | 0.156 | **0.000848** | 0.000951 |
| 0.4 | 22.025 | 0.775 | 0.158 | 0.000849 | **0.000926** |
| 0.5 | 21.927 | 0.773 | 0.163 | 0.000871 | 0.000982 |
| 0.6 | 21.793 | 0.768 | 0.170 | **0.000848** | 0.000978 |

Table 4: Comparison of different rendering strategies for Binocular3DGS trained with dropout probability $p = 0.2$ on the LLFF dataset. Baseline: Training and Rendering without dropout. A: Single random dropout rendering. B: Averaging multiple random dropout renderings (five times). C: Using all Gaussians with opacity scaled by $(1 - p)$.

| Strategy | PSNR ↑ | SSIM ↑ | LPIPS ↓ | Train CA ↓ | Test CA ↓ |
|---|---|---|---|---|---|
| Baseline | 21.440 | 0.751 | 0.168 | 0.001845 | 0.001951 |
| A | 21.977 | 0.769 | 0.162 | 0.000875 | 0.000978 |
| B | **22.124** | 0.776 | 0.157 | 0.000875 | 0.000978 |
| C | 22.123 | **0.777** | **0.154** | 0.000875 | 0.000978 |

only a slight increase remains. In addition, both strategies improve PSNR over the baseline, showing their effectiveness in stabilizing training and enhancing reconstruction quality.

**(2) Ablation on Dropout Probability $p$ and Rendering Strategies.** Table 3 presents the ablation study on different dropout probabilities $p$. We observe that setting $p$ to 0.2 achieves the best overall reconstruction quality metrics. Although this setting yields the lowest training-view CA score, the testing-view CA is not minimized, suggesting that CA score is not strictly monotonic with reconstruction quality. Extremely low CA scores do not necessarily guarantee better rendering results once a certain threshold is reached. In addition, we evaluate three rendering strategies under the same dropout-trained model in Table 4. Strategy A applies random dropout at inference with probability $p$. Strategy B performs multiple stochastic dropout renderings and averages the results. Strategy C scales the opacity of all Gaussians by $(1 - p)$ and renders the image in a single pass. Our experiments show that Strategy C achieves the best quality-efficiency trade-off with single-pass rendering, while Strategy B performs similarly but is five times slower.

**(3) Ablation on Opacity Noise Scale $\sigma$.** Table 5 reports the ablation study on different opacity noise scales $\sigma$. We observe that $\sigma = 0.8$ achieves the best reconstruction quality. Increasing $\sigma$ gradually reduces both training and testing CA scores. However, when $\sigma$ goes below 0.8, the reconstruction quality starts to degrade, despite further reductions in CA. This again confirms that lower CA does not always improve rendering quality beyond a certain point.

Table 5: Ablation study on the noise scale $\sigma$ applied to opacity on the LLFF dataset.

| $\sigma$ (Noise Scale) | PSNR ↑ | SSIM ↑ | LPIPS ↓ | Train CA ↓ | Test CA ↓ |
|---|---|---|---|---|---|
| 0.0 (Baseline) | 21.440 | 0.751 | 0.168 | 0.001845 | 0.001951 |
| 0.1 | 21.647 | 0.757 | 0.165 | 0.001409 | 0.001511 |
| 0.2 | 21.864 | 0.764 | 0.161 | 0.001126 | 0.001239 |
| 0.3 | 21.942 | 0.769 | 0.158 | 0.000959 | 0.001074 |
| 0.4 | 22.065 | 0.774 | 0.155 | 0.000859 | 0.000964 |
| 0.5 | 22.072 | 0.776 | **0.154** | 0.000861 | 0.000968 |
| 0.6 | 21.999 | 0.777 | 0.155 | 0.000794 | 0.000895 |
| 0.7 | 22.076 | 0.777 | 0.155 | 0.000712 | 0.000798 |
| 0.8 | **22.119** | **0.780** | 0.155 | 0.000660 | 0.000762 |
| 0.9 | 22.059 | 0.779 | 0.157 | 0.000589 | 0.000693 |
| 1.0 | 22.053 | 0.779 | 0.159 | **0.000560** | **0.000640** |

## 5    Conclusion

In this paper, we introduce the concept of co-adaptation in 3D Gaussian Splatting (3DGS) [1] and reveal its link to appearance artifacts under sparse-view settings. We propose the Co-Adaptation Score (CA) metric to quantify this effect and show that higher view density naturally reduces co-adaptation. Motivated by this, we propose two simple strategies—dropout regularization and opacity noise injection—that effectively mitigate co-adaptation and improve sparse-view novel view synthesis. We hope our findings inspire future work on mitigating co-adaptation in learned 3D representations.

# 6 Acknowledgement

This work was supported by the Shenzhen Science and Technology Project under Grant (JCYJ20220818101001004, KJZD20240903103210014).

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

# A  Appendix / Supplemental Material

## A.1  Theoretical Analysis of Co-Adaptation Score

**Note.** In the main paper, CA is reported as the average pixel-wise variance over the commonly visible region of an entire image. Here, for clarity, we derive the formula for a single pixel $u$.

We begin by formalizing how random dropout perturbs the rendered color. Under a dropout mask $\mathbf{z} = (z_1, \ldots, z_n)$—where each $z_i \in \{0, 1\}$ independently indicates whether Gaussian $i$ is kept—the color at pixel $u$ is

$$C^{(\mathbf{z})}(u) = \sum_{i \in \mathcal{N}(u)} z_i\, c_i\, \alpha_i \prod_{j<i}\bigl(1 - z_j\, \alpha_j\bigr). \tag{7}$$

Here $\mathcal{N}(u)$ is the depth-sorted list of Gaussians projecting to pixel $u$, $c_i$ and $\alpha_i$ are the color and opacity of Gaussian $i$, and the product term is the usual front-to-back transmittance. The Co-Adaptation Score measures how much $C^{(\mathbf{z})}(u)$ fluctuates as we sample different masks:

$$\mathrm{CA}(u) = \mathrm{Var}_{\mathbf{z}}\bigl(C^{(\mathbf{z})}(u)\bigr). \tag{8}$$

A large variance indicates that dropping a subset of Gaussians greatly alters the pixel color, i.e. that Gaussians have become overly co-dependent. To see how CA depends on the individual parameters, we apply a first-order expansion of the transmittance:

$$\prod_{j<i}(1 - z_j\alpha_j) \approx 1 - \sum_{j<i} z_j\, \alpha_j. \tag{9}$$

Substituting back and discarding second-order (and higher) interactions yields

$$C^{(\mathbf{z})}(u) \approx \sum_i z_i\, c_i\, \alpha_i \; - \; \sum_i z_i\, c_i\, \alpha_i \sum_{j<i} z_j\, \alpha_j. \tag{10}$$

In this approximation, the dominant term is the simple sum of active contributions. Neglecting the smaller second term, the variance becomes

$$\mathrm{CA}(u) \approx \mathrm{Var}_{\mathbf{z}}\Bigl(\sum_i z_i\, c_i\, \alpha_i\Bigr). \tag{11}$$

Since each $z_i$ is a Bernoulli random variable with success probability $\Pr[z_i = 1] = 1 - p$, where $p$ denotes the dropout probability, its first and second moments are given by

$$\mathbb{E}[z_i] = 1 - p, \quad \mathrm{Var}(z_i) = p(1 - p). \tag{12}$$

As the dropout decisions $z_i$ are independent across different Gaussians, we can derive the variance of the total weighted sum of Gaussian contributions to pixel $u$ as follows:

$$\mathrm{Var}\left(\sum_i z_i\, c_i\, \alpha_i\right) = \sum_i \mathrm{Var}(z_i)\,(c_i\, \alpha_i)^2 = p(1 - p)\sum_i (c_i\, \alpha_i)^2. \tag{13}$$

Hence, we obtain a simplified expression for the Co-Adaptation Score (CA) at pixel $u$:

$$\mathrm{CA}(u) \approx p(1 - p)\sum_{i \in \mathcal{N}(u)} (c_i\, \alpha_i)^2. \tag{14}$$

This final form shows that the CA score grows proportionally with the sum of squared weighted color-opacity terms across all Gaussians contributing to pixel $u$, scaled by the dropout-related variance factor $p(1 - p)$. It quantitatively captures how variations in both color and opacity among Gaussians manifest as co-adaptation under random dropout perturbations.

Equation 14 highlights several key insights: (1) Dependence on opacity and color: Gaussians with large $c_i\alpha_i$ terms contribute more significantly to the CA score. When these contributions are similar or redundant, the resulting variance—and thus co-adaptation—is low. (2) Effect of dropout rate: The multiplicative factor $p(1-p)$ reaches its maximum at $p = 0.5$, justifying our choice to adopt this value in practice to enhance the sensitivity of the CA metric. (3) Regularization implications: Perturbations to $c_i\alpha_i$ (e.g., via noise injection or dropout) reduce the magnitudes of the summation terms, leading to lower CA scores and implicitly mitigating excessive interdependence among Gaussians.

In summary, this theoretical analysis substantiates the CA score as a principled indicator of color-opacity coupling strength under dropout perturbations, thereby providing a foundation for the design of our co-adaptation suppression strategies.

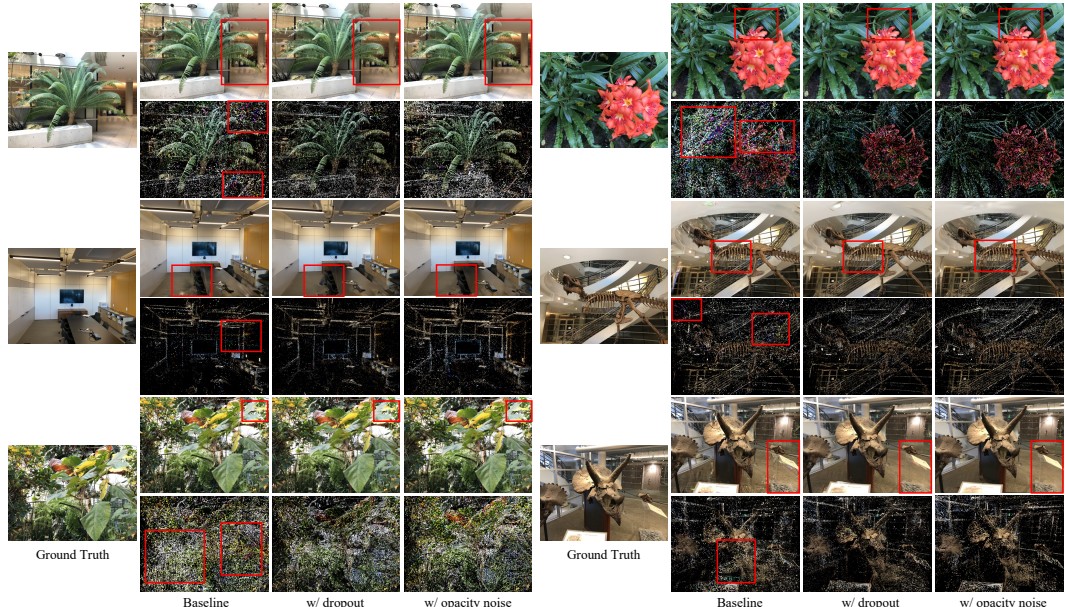

Figure 9: Visual comparison on the LLFF dataset based on 3DGS. Suppressing co-adaptation reduces color noise (see the changes in the GS point cloud) and improves scene geometry and detail quality.

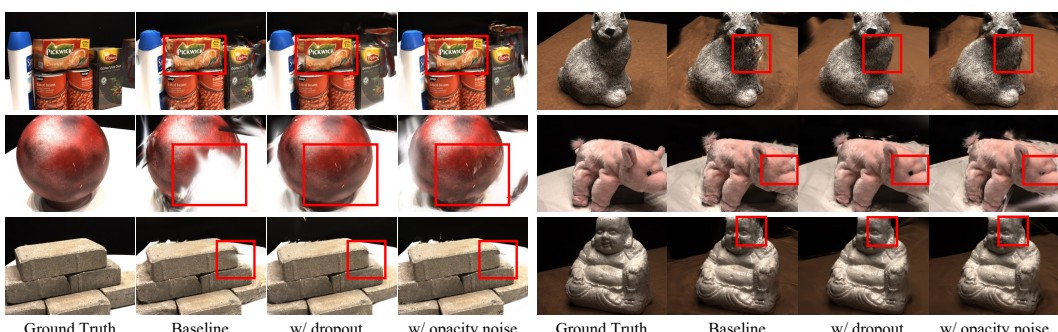

Figure 10: Visual comparison on the DTU dataset based on 3DGS. Suppressing co-adaptation leads to more consistent scene fusion across views, resulting in clearer structure and more accurate details.

## A.2 Extended Qualitative Results

We present additional qualitative results on the LLFF [47], DTU [51], and Blender [52] datasets to further illustrate the visual improvements brought by our co-adaptation suppression strategies.

**LLFF.** We first examine the LLFF dataset, where improvements are clearly observed both in rendered images and Gaussian point clouds. As shown in Figure 9, our methods yield point clouds that are cleaner and more geometrically coherent. The baseline 3D Gaussian Splatting (3DGS) [1] often produces scattered, colorful speckles, which are significantly suppressed after applying our co-adaptation regularization. These visual improvements in point clouds directly lead to better rendering in novel views. For instance, in the *trex* scene, fossil bones exhibit high-frequency flickering and speckling in the baseline, which are clearly alleviated with our methods. In the *flower* scene, petals and leaves display incorrect colorization under novel views, while the regularized versions better preserve the natural appearance. Similarly, in the *fern* scene, severe color inconsistencies appear on the pole, ground, and leaf textures under the baseline, but these are significantly reduced when co-adaptation is suppressed. Overall, the results suggest that reducing co-adaptation enhances both structural consistency and appearance fidelity in sparse-view 3DGS.

**DTU.** On the DTU dataset, we observe that both co-adaptation suppression strategies significantly enhance structural coherence and reduce visual artifacts in sparse-view 3DGS. As illustrated in

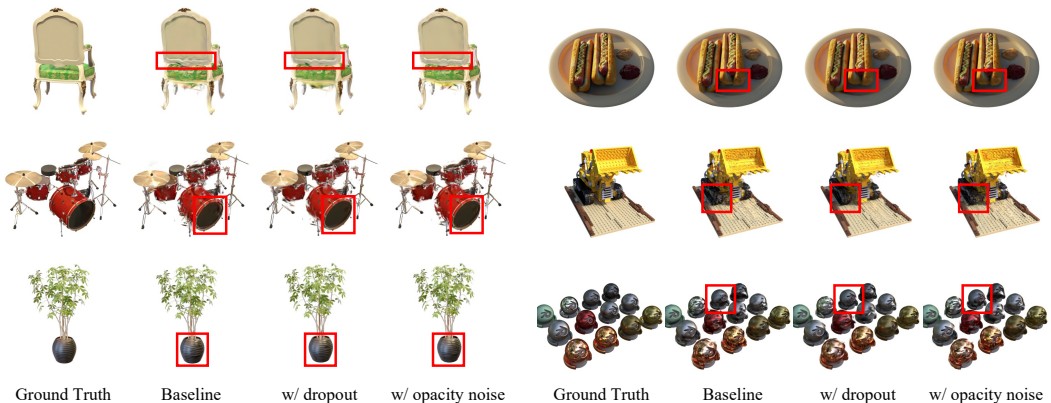

| Ground Truth | Baseline | w/ dropout | w/ opacity noise | Ground Truth | Baseline | w/ dropout | w/ opacity noise |

Figure 11: Visual comparison on the Blender dataset based on 3DGS. Suppressing co-adaptation reduces floating artifacts and enhances the completeness and clarity of fine structures.

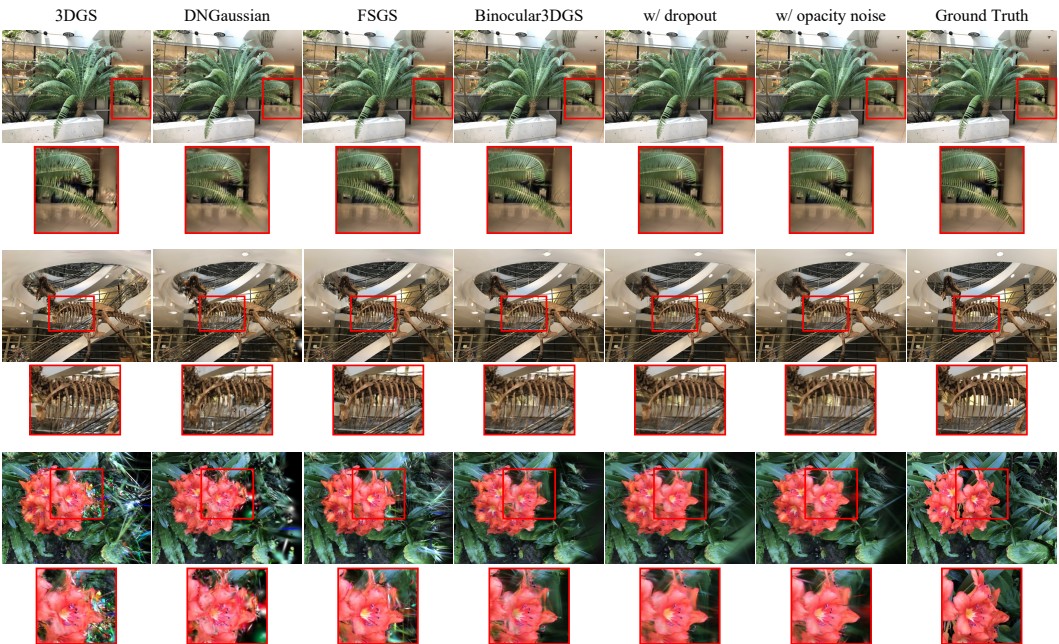

| 3DGS | DNGaussian | FSGS | Binocular3DGS | w/ dropout | w/ opacity noise | Ground Truth |

Figure 12: Visual comparison on LLFF dataset across multiple methods, using Binocular3DGS as the baseline. Suppressing co-adaptation further reduces colorful speckle artifacts and floating noise, while enhancing the completeness and structural coherence of the reconstructed geometry.

Figure 10, suppressing co-adaptation leads to improved fusion across views, enabling more accurate and consistent detail reconstruction. In the scene containing multiple stacked boxes, our methods help preserve fine-scale textual details on the box surfaces, with the characters on the packaging appearing noticeably clearer compared to the baseline. In the scene with the red ball, our regularization strategies substantially reduce the bright floating speckles that frequently appear in front of the camera, especially near the lens center, leading to a cleaner and more complete object reconstruction. Furthermore, in the scene featuring a small pink pig plush toy, the eyes of the pig—barely visible in the baseline—are successfully reconstructed under novel views when using either of our suppression methods. These improvements confirm that controlling co-adaptation strengthens the model's ability to render fine geometric and appearance details in complex real-world scenes.

**Blender.** On the Blender dataset, we observe consistent improvements in rendering quality and structural fidelity when applying co-adaptation suppression strategies. As shown in Figure 11, both dropout and opacity noise help reduce subtle artifacts and enhance reconstruction realism under sparse-view supervision. In the *chair* and *drums* scenes, novel view renderings generated with our methods

Table 6: Ablation study on the noise scale $\sigma$ applied to **scale parameters** in Binocular3DGS on the LLFF dataset. We use multiplicative noise with different $\sigma$ values and evaluate the impact on rendering quality and co-adaptation.

| Scale Noise $\sigma$ | PSNR ↑ | SSIM ↑ | LPIPS ↓ | Train CA ↓ | Test CA ↓ |
|---|---|---|---|---|---|
| 0.0 (Baseline) | 21.440 | 0.751 | 0.168 | 0.001845 | 0.001951 |
| 0.1 | 21.551 | 0.755 | 0.166 | 0.001520 | 0.001682 |
| 0.2 | **21.760** | 0.760 | 0.161 | 0.001326 | 0.001432 |
| 0.3 | 21.724 | 0.762 | 0.161 | 0.001148 | 0.001238 |
| 0.4 | 21.690 | 0.762 | 0.161 | 0.001075 | 0.001237 |
| 0.5 | 21.674 | 0.765 | **0.159** | 0.001009 | 0.001091 |
| 0.6 | 21.685 | **0.766** | **0.159** | **0.000927** | 0.001014 |
| 0.7 | 21.597 | 0.762 | 0.162 | 0.000945 | **0.001012** |

exhibit fewer detail-related artifacts, particularly around fine structures such as chair legs and drum edges, resulting in more accurate geometry and cleaner appearance. In the *ficus* scene, both strategies significantly improve the photorealism of the reconstructed flowerpot, leading to more consistent shading and geometry that better match the true scene layout. Additionally, in the *hotdog* scene, we observe a clear reduction in floating speckle artifacts outside the object boundary—especially near the sausage and tray—demonstrating the effectiveness of co-adaptation suppression in eliminating non-semantic visual noise. Together, these results highlight the generalizability of our techniques to synthetic datasets with complex textures and fine-scale geometry.

**Method Comparison.** Figure 12 presents a visual comparison across multiple 3DGS methods, using Binocular3DGS [10] as the baseline. Binocular3DGS is a strong baseline with geometry-aware initialization, yet our co-adaptation suppression strategies (dropout and opacity noise) still lead to noticeable improvements. In the *flower* scene, existing sparse-view approaches exhibit occasional color speckles and geometry gaps; applying our methods further reduces these artifacts and yields more complete, coherent novel view reconstructions.

## A.3 Exploration of Noise Injection on Different Gaussian Attributes

Beyond the opacity noise experiments presented in the main paper, we further explore noise injection on other Gaussian parameters, including the scale, position, and SH (spherical harmonics) coefficients. These experiments aim to analyze whether perturbing other parameters can also suppress co-adaptation and improve rendering quality under sparse-view settings.

**Scale Noise Injection.** We first examine the effect of injecting multiplicative noise into the scale parameters of Gaussians. Similar to opacity noise, we apply multiplicative perturbation as follows:

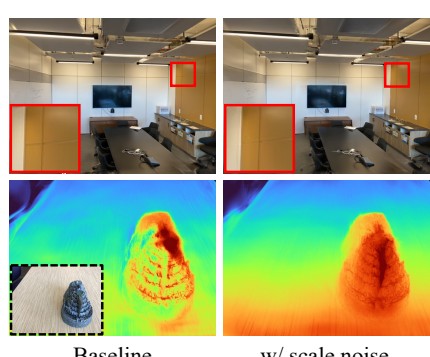

$$\tilde{s} = s \cdot (1 + \epsilon), \quad \epsilon \sim \mathcal{N}(0, \sigma^2), \qquad (15)$$

where $s$ denotes the original scale and $\sigma$ is the noise strength. The results of this experiment are summarized in Table 6. Compared to the Binocular3DGS baseline, scale noise yields limited improvements in overall reconstruction metrics such as PSNR, SSIM [56], and LPIPS [57]. However, it reduces the co-adaptation strength (CA) on both training and novel views, indicating that scale perturbation helps decouple Gaussian interactions to some extent. This finding aligns with our visual observations in Figure 13, where scale noise improves scene-level geo-

Baseline          w/ scale noise

Figure 13: Visual comparison of scale noise effects in Binocular3DGS.

metric coherence in some cases, but introduces mild local blurring in object details. The observed trade-off can be explained by the functional role of scale in Gaussian rendering: perturbing scale slightly alters the spatial support of Gaussians, which may improve coverage and fusion in sparse-view settings, but risks oversmoothing fine structures. As a result, while scale noise contributes to better scene fusion in some examples, it reduces local detail sharpness, leading to modest overall gains.

**Position Noise Injection.** For position parameters, we introduce additive noise scaled by the nearest-neighbor distance of each Gaussian, which perturbs their spatial locations and disrupts alignment

Table 7: Ablation study of **position** and **SH noise** injection on the LLFF dataset. Position noise is added as Gaussian noise scaled by nearest-neighbor distance; SH noise is applied multiplicatively.

| Noise Scale | PSNR ↑ | SSIM ↑ | LPIPS ↓ | Train CA ↓ | Test CA ↓ |
|---|---|---|---|---|---|
| 0.0 (Baseline) | 21.440 | 0.751 | 0.168 | 0.001845 | 0.001951 |
| **Position Noise** (additive) | | | | | |
| 0.1 | 21.093 | 0.742 | 0.171 | 0.001701 | 0.001812 |
| 0.3 | 20.760 | 0.726 | 0.182 | 0.001594 | 0.001730 |
| 0.5 | 20.540 | 0.715 | 0.189 | 0.001653 | 0.001733 |
| **SH Noise** (multiplicative) | | | | | |
| 0.1 | 21.475 | 0.749 | 0.171 | 0.001769 | 0.001876 |
| 0.3 | 21.391 | 0.750 | 0.170 | 0.001633 | 0.001753 |
| 0.5 | 21.390 | 0.751 | 0.168 | 0.001799 | 0.001906 |

Table 8: Ablation study on different **SH orders** used in 3DGS under sparse-view settings on the LLFF dataset. We evaluate rendering quality and co-adaptation across SH0 to SH3.

| SH Order | PSNR ↑ | SSIM ↑ | LPIPS ↓ | Train CA ↓ | Test CA ↓ |
|---|---|---|---|---|---|
| 0 | 19.293 | 0.652 | 0.235 | **0.006736** | **0.007259** |
| 1 | 19.239 | 0.651 | 0.234 | 0.007087 | 0.007587 |
| 2 | **19.377** | **0.657** | **0.229** | 0.007167 | 0.007858 |
| 3 (Baseline) | 19.357 | 0.651 | 0.232 | 0.007543 | 0.008206 |

with the image plane. Even under small perturbations, this leads to degraded convergence and blurred geometry in sparse-view training. Although a coarse-to-fine training strategy can somewhat stabilize optimization, its effect on overall reconstruction quality remains limited.

**SH Noise Injection.** For SH parameters, we apply multiplicative noise in the form $\tilde{c} = c \cdot (1 + \epsilon)$ with $\epsilon \sim \mathcal{N}(0, \sigma^2)$ to perturb the directional color components. However, since SH parameters do not affect the spatial configuration of Gaussians along a ray, they do not influence the contribution weights of Gaussians during the alpha blending process. As a result, SH noise has limited effect on alleviating co-adaptation. While it may seem intuitive that perturbing SH and opacity could yield similar regularization effects—since both influence final pixel colors—their impact on rendering differs fundamentally. Opacity noise directly modifies the weight assigned to each Gaussian during alpha blending, thereby altering the effective contribution and even the number of Gaussians used to render a pixel. This enables opacity noise to disrupt the co-adapted structure more substantially. In contrast, SH perturbation only changes the emitted color of each Gaussian without modifying its blending weight, thus providing limited relief against co-adaptation.

## A.4 Extended Ablation on SH Orders

To investigate whether the choice of SH (spherical harmonics) order plays a significant role in co-adaptation, we conduct a controlled ablation study using SH orders from 0 to 3 in vanilla 3DGS training under sparse-view settings. The quantitative results are shown in Table 8, and corresponding qualitative comparisons of Gaussian point clouds are visualized in Figure 14.

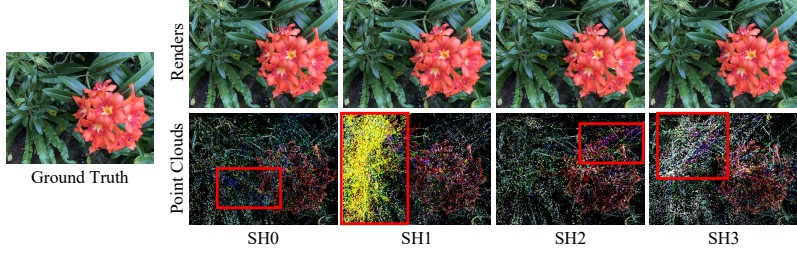

Figure 14: Visualization of Gaussian point clouds under different SH orders on the LLFF *flower* scene. SH0 yields cleaner points, while higher SH orders occasionally introduce more color speckles.

We observe that lower SH orders, especially SH0, tend to yield slightly lower co-adaptation scores and cleaner Gaussian point distributions. In contrast, SH1 to SH3 occasionally produce localized color speckles, but these artifacts appear stochastically—without a consistent correlation to SH order. For example, SH1 may exhibit more noticeable artifacts than SH2 in certain training runs. Overall, the differences in co-adaptation strength across SH0 to SH3 remain relatively minor, and the emergence of appearance artifacts is not strongly determined by the SH order. These findings suggest that while SH complexity may influence directional appearance expressiveness, it is not a dominant factor in controlling co-adaptation or mitigating color artifacts in sparse-view 3DGS.

### A.5  Gaussian Count Comparison

To supplement our main experiments, we provide a detailed comparison of the final average number of Gaussians under different training strategies on the LLFF dataset. Specifically, we compare vanilla 3DGS and Binocular3DGS baselines with our proposed dropout and opacity noise regularization strategies. Results are summarized in Table 9.

Table 9: Final Gaussian count and performance metrics under different strategies on LLFF.

| Setting | PSNR ↑ | SSIM ↑ | LPIPS ↓ | Train CA ↓ | Test CA ↓ | Final GS Count |
|---|---|---|---|---|---|---|
| 3DGS [1] | 19.36 | 0.651 | 0.232 | 0.007543 | 0.008206 | 111137 |
| w/ dropout | 20.20 | 0.691 | 0.211 | 0.001752 | 0.002340 | 115046 |
| w/ opacity noise | 19.91 | 0.676 | 0.223 | 0.001531 | 0.002300 | 98356 |
| Binocular3DGS [10] | 21.44 | 0.751 | 0.168 | 0.001845 | 0.001951 | 102021 |
| w/ dropout | 22.12 | 0.777 | 0.154 | 0.000875 | 0.000978 | 122340 |
| w/ opacity noise | 22.12 | 0.780 | 0.155 | 0.000660 | 0.000762 | 104832 |

These results show that our method does not improve the CA-score by duplicating Gaussians. The final Gaussian count remains comparable to, or even lower than, the vanilla baselines. This supports that the improvements in CA metrics reflect genuine regularization effects on opacity and color distributions, rather than a degenerate collapse of the representation space.

### A.6  Exploration of a Color Variance Metric (CV)

To further understand how our proposed dropout and opacity noise strategies influence color distribution, we introduce an auxiliary metric to measure the variance of rendered colors while decoupling it from opacity variations. Specifically, we adopt a pixel-wise color variance formulation that weighs contributing colors $c_i$ using their rendering compositing weights $w_i$, defined as:

$$w_i = \alpha_i \cdot \prod_{j=1}^{i-1}(1 - \alpha_j) \tag{16}$$

We compute the color variance of each pixel using the following formulation:

$$\mathrm{Var}(c) = \frac{\sum_i w_i c_i^2}{\sum_i w_i} - \left(\frac{\sum_i w_i c_i}{\sum_i w_i}\right)^2 \tag{17}$$

This formulation reflects the variance of color contributions that is decoupled from the opacity aggregation behavior during splatting. Averaging the per-pixel variance over the entire image yields a global **color variance metric (CV)**.

Table 10 reports the CV metric, co-adaptation scores (CA), and final Gaussian statistics under different regularization strategies. The experiments are conducted on the LLFF dataset using both the vanilla 3DGS and Binocular3DGS baseline.

Table 10: **Color variance (CV)** and co-adaptation score (CA) under different strategies on LLFF.

| Setting | PSNR ↑ | Train CA ↓ | Test CA ↓ | Train CV ↓ | Test CV ↓ | Final GS Count | GS Radius |
|---|---|---|---|---|---|---|---|
| Binocular3DGS [10] | 21.44 | 0.001845 | 0.001951 | 0.2496 | 0.2563 | 102021 | 21.38 |
| w/ dropout | 22.12 | 0.000875 | 0.000978 | 0.0992 | 0.1065 | 122340 | 35.93 |
| w/ opacity noise | 22.12 | 0.000660 | 0.000762 | 0.0476 | 0.0534 | 104832 | 31.64 |

We observe that both dropout and opacity noise substantially reduce the proposed CV metric and the CA score. The opacity noise achieves the lowest CV. In contrast, dropout occasionally results in slightly larger Gaussians, which may contribute to a marginally higher CV and CA.

Additionally, we have also experimented with an alternative regularization scheme that explicitly aligns the color directions of Gaussians (after normalization) with the ground truth ray direction. However, this strict constraint significantly degraded rendering quality, likely due to the loss of flexibility in 3D Gaussian expression.

## A.7  Comparison with Alternative Strategies for Reducing Co-Adaptation

To better situate our proposed methods within the broader landscape of appearance artifact mitigation and generalization enhancement in sparse-view 3DGS, we additionally compare with three alternative strategies that are known or expected to influence model co-adaptation behavior. All comparisons are conducted based on the Binocular3DGS baseline and evaluated on the LLFF dataset.

- **Baseline w/ Scale Noise:** We inject multiplicative noise into the scale parameters of Gaussians during training. This strategy introduces spatial variation and slightly perturbs the scale consistency across views, which can act as a regularizer against overfitting and co-adaptation.

- **Baseline w/ AbsGS [58]:** We apply pixel-wise absolute gradient accumulation for Gaussian densification, replacing the standard gradient used in vanilla 3DGS. This method promotes the preservation of fine structures and reduces appearance artifacts, which may help alleviate view-specific co-adaptation.

- **Baseline w/o Opacity Decay [10]:** We remove the default multiplicative decay applied to Gaussian opacity parameters ($\times 0.995$ per iteration). Without this decay, redundant Gaussian opacities tend to maintain, causing higher co-adaptation scores, reduced sparsity, and more pronounced floating artifacts.

Table 11 summarizes the comparison results across several quantitative metrics, including rendering quality (PSNR, SSIM, LPIPS) and co-adaptation strength (Train/Test CA).

Table 11: Comparison of different strategies for mitigating co-adaptation and improving rendering quality on LLFF.

| Setting | PSNR ↑ | SSIM ↑ | LPIPS ↓ | Train CA ↓ | Test CA ↓ |
|---|---|---|---|---|---|
| baseline [10] | 21.44 | 0.751 | 0.168 | 0.001845 | 0.001951 |
| w/ dropout | 22.12 | 0.777 | 0.154 | 0.000875 | 0.000978 |
| w/ opacity noise | 22.12 | 0.780 | 0.155 | 0.000660 | 0.000762 |
| w/ scale noise | 21.68 | 0.766 | 0.159 | 0.000927 | 0.001014 |
| w/ AbsGS [58] | 21.61 | 0.754 | 0.163 | 0.001446 | 0.001593 |
| w/o opacity decay [10] | 20.26 | 0.708 | 0.202 | 0.004413 | 0.004830 |

We observe that dropout and opacity noise yield the lowest co-adaptation scores and the best rendering metrics overall. Scale noise also helps mitigate co-adaptation to some extent, while AbsGS provides a balance between detail preservation and regularization. In contrast, removing the opacity decay results in a substantial increase in co-adaptation scores and degrades rendering quality.

These results provide empirical evidence that multiple strategies—both architectural and regularization-based—can influence co-adaptation behavior in 3DGS. Our proposed dropout and opacity noise methods remain the most effective among them in reducing co-adaptation.

## A.8  Impact on Geometry Reconstruction

Beyond the improvement in appearance fidelity, we also assess whether our proposed regularization strategies enhance the quality of geometry reconstruction in sparse-view settings. To this end, we compare the predicted depth maps from trained 3DGS models against pseudo ground-truth depth maps gained under an 18-view denser configuration on the LLFF dataset.

We evaluate the quality of depth using the following standard metrics:

- AbsRel: Mean absolute relative error between predicted and reference depths.
- RMSE: Root mean squared error, capturing the overall deviation.
- MAE: Mean absolute error, measuring absolute differences in depth.
- Log10: Logarithmic error, which is more sensitive to errors in distant regions.

Table 12: Depth quality under sparse-view settings (LLFF, 3 views).

| Setting | AbsRel ↓ | RMSE ↓ | MAE ↓ | Log10 ↓ |
|---|---|---|---|---|
| baseline [2] | 0.0732 | 4.2785 | 2.5802 | 0.0314 |
| w/ dropout | 0.0676 | 4.1048 | 2.4258 | 0.0295 |
| w/ opacity noise | 0.0651 | 3.9182 | 2.2923 | 0.0276 |

We observe that both dropout and opacity noise consistently improve geometry prediction quality across all four metrics, with opacity noise showing the strongest impact.

### A.9 Exploration of Advanced Dropout Strategies

To further investigate how to suppress co-adaptation more effectively in sparse-view 3DGS reconstruction, we explore several targeted dropout strategies beyond uniform random dropout. All experiments are conducted using the LLFF dataset and the Binocular3DGS baseline. Below, we summarize the design and evaluation of three dropout variants.

**Concrete Dropout.** We implement a variant of Concrete Dropout [59] by learning a per-Gaussian dropout probability $p_i \in (0, 1)$. During training, we sample a soft mask $z_i$ using the Binary Concrete distribution:

$$z_i = \text{Sigmoid}\left(\frac{\log(p_i) - \log(1 - p_i) + \log(u_i) - \log(1 - u_i)}{\tau}\right), \quad u_i \sim \mathcal{U}(0, 1) \quad (18)$$

where $\tau = 0.1$ is a temperature hyperparameter. The final Gaussian opacity is updated as $o_i \leftarrow o_i \cdot (1 - z_i)$.

This method allows end-to-end learning of dropout uncertainty. However, under sparse-view settings, we find that the soft mask introduces insufficient regularization, resulting in significant degradation.

**Density-based Dropout.** We hypothesize that Gaussians in dense spatial regions are more likely to over co-adapt. Hence, we compute nearest-neighbor density scores and assign dropout probabilities proportionally. Specifically, we assign higher dropout rates to denser regions and lower rates to sparser ones. Dropout rates are linearly mapped from 0.2 (sparsest) to 0.5 (densest).

Although this strategy introduces a more selective regularization mechanism and achieves noticeable improvement over the baseline, it leads to a larger number of remaining Gaussians compared to random dropout. Moreover, it does not yield a clear advantage in rendering quality and in some cases even underperforms the uniform random dropout strategy.

**Geometry-aware Dropout.** To account for high-order geometric co-adaptation, we estimate local geometric complexity by computing structural variation among each Gaussian's nearest neighbors. However, the computational cost of this strategy—approximately $\mathcal{O}(N^3)$ for $N$ Gaussians—renders it infeasible in practice. Thus, we omit this strategy from ablation studies.

Table 13: Comparison of dropout strategies on LLFF using the Binocular3DGS baseline.

| Setting | PSNR ↑ | SSIM ↑ | LPIPS ↓ | Train CA ↓ | Test CA ↓ | Final GS Count |
|---|---|---|---|---|---|---|
| baseline | 21.44 | 0.751 | 0.168 | 0.001845 | 0.001951 | 102021 |
| w/ random dropout | 22.12 | 0.777 | 0.154 | 0.000875 | 0.000978 | 122340 |
| w/ concrete dropout | 20.88 | 0.731 | 0.205 | 0.004680 | 0.005320 | 22959 |
| w/ density-based dropout | 22.05 | 0.773 | 0.158 | 0.000478 | 0.000536 | 150968 |
| w/ geometry-aware dropout | Too costly; computational complexity estimated as $\mathcal{O}(N^3)$ | | | | | |

As shown in Table 13, while concrete and density-based dropout offer additional perspectives on targeted regularization, our experiments show that uniform random dropout remains the most effective

and practical solution under sparse-view conditions. Geometry-aware strategies may hold future potential, but their high computational overhead limits immediate applicability.

## A.10 Exploration of Multi-View Learning

To further explore the impact of multi-view learning under sparse-view 3DGS settings, we implemented the official MVGS [60] using 3 input views. As shown in Table 14, the MVGS model yields significantly degraded performance across all image quality metrics under sparse-view input, suggesting its lack of adaptation to sparse supervision.

Table 14: Sparse-view MVGS performance on LLFF.

| Method | PSNR ↑ | SSIM ↑ | LPIPS ↓ |
|---|---|---|---|
| MVGS [60] | 15.45 | 0.507 | 0.362 |

We further added a simple multi-view training module on top of vanilla 3DGS, synchronizing three views per iteration (MV=3). This accelerates convergence due to stronger multi-view gradients, but also results in aggressive Gaussian densification and a sharp rise in co-adaptation, as indicated by the co-adaptation scores.

We assess this integration under different training iterations (1k, 3k, 5k, 10k), as shown in Table 15.

Table 15: Impact of multi-view (MV=3) learning on sparse-view 3DGS under different training iterations.

| Setting | PSNR ↑ | SSIM ↑ | LPIPS ↓ | Train CA ↓ | Test CA ↓ | GS Count (Avg.) |
|---|---|---|---|---|---|---|
| 3DGS (10k) | 19.36 | 0.651 | 0.232 | 0.007543 | 0.008206 | 111137 |
| w/ MV (10k) | 18.75 | 0.639 | 0.245 | 0.016841 | 0.024286 | 1144360 |
| w/ MV (5k) | 18.97 | 0.646 | 0.240 | 0.018252 | 0.022790 | 635671 |
| w/ MV (3k) | 19.41 | 0.665 | 0.228 | 0.011994 | 0.013679 | 259014 |
| w/ MV (1k) | 19.55 | 0.656 | 0.292 | 0.008352 | 0.008635 | 22910 |

While multi-view supervision introduces stronger signals and faster convergence, it does not consistently reduce co-adaptation scores or sparse-view artifacts. These findings suggest that multi-view strategies require additional regularization mechanisms to be effective under sparse-view reconstruction settings. Nevertheless, the direction remains promising for future exploration.

## A.11 Extended Experimental Details

To ensure fair comparisons, we adopt unified parameter settings across all scenes within each dataset. For example, while the original DNGaussian [2] uses different training configurations for each Blender scene, we apply a consistent setup across all scenes. Binocular3DGS is retrained using white backgrounds to match other methods, as its original results were obtained using black backgrounds. For co-adaptation suppression, we use a fixed dropout probability of 0.2 across all methods and datasets, based on ablations conducted on Binocular3DGS. Because each method learns a distinct opacity distribution, we tune the opacity noise scale individually for each method within $[0.05, 0.8]$ and fix it across all scenes in the same dataset. For CA score computation, we randomly drop 50% of the contributing Gaussians for each view and calculate the pixel-wise variance across multiple renderings. The number of training iterations for each baseline follows its official implementation.

## A.12 Future Work and Broader Impact

While our study primarily focuses on quantifying and alleviating co-adaptation in 3D Gaussian Splatting (3DGS), the underlying analysis and methodology hold broader implications for 3D representation learning. The core idea of leveraging dropout-inspired regularization provides a simple yet powerful lens for understanding and mitigating undesirable dependencies among Gaussian primitives.

Specifically, our findings suggest several promising directions for future exploration:

- **Point Initialization and Densification.** The quantification of co-adaptation may inspire new initialization or adaptive densification strategies that enhance convergence stability in sparse-view 3DGS training.
- **Per-Gaussian Feature Learning.** Extending co-adaptation analysis to learn better per-Gaussian semantic or appearance embeddings could improve tasks such as 3D segmentation, object retrieval, or open-vocabulary scene understanding [28, 61, 62, 63, 64].
- **Generalizable 3D Representations.** The insight into co-adaptation may help build 3D representations that generalize better across novel views, modalities, or domains, thus benefiting reconstruction, editing, and generation tasks.
- **Adaptive and Intelligent Regularization Strategies.** Building upon our exploration of dropout-based mechanisms, future work could investigate more targeted and adaptive strategies for better co-adaptation alleviating and novel view synthesis generalizability.

In summary, while dropout serves as the conceptual inspiration, our work goes beyond a direct adaptation. By uncovering and formalizing the co-adaptation phenomenon in 3DGS, we offer a new perspective that may benefit a wide range of point-based 3D learning frameworks. We hope this direction will stimulate further research on robust, interpretable, and generalizable 3D representations.

## A.13 Limitations

Although our proposed strategies—random Gaussian dropout and opacity noise injection—effectively reduce co-adaptation and enhance novel view synthesis under sparse supervision, we observe that overly suppressing co-adaptation does not always yield further improvements in rendering quality. As evidenced in our ablation studies, rendering performance tends to plateau or even degrade once co-adaptation scores fall below a certain threshold. This suggests that co-adaptation is not inherently harmful; rather, it constitutes an essential aspect of 3DGS, facilitating cooperation among Gaussians to model fine-grained appearance and geometry. The negative effects of co-adaptation primarily emerge under sparse-view training, where excessive entanglement can lead to artifacts or overfitting. Consequently, while moderate suppression is beneficial for improving generalization, excessive suppression may compromise the model's capacity to accurately fit scene content, ultimately limiting expressiveness. Our goal is thus not to eliminate co-adaptation entirely, but to selectively mitigate its detrimental forms. Future work may consider adaptive suppression mechanisms that dynamically balance generalization and representational fidelity.

