# OpenReview forum: "Quantifying and Alleviating Co-Adaptation in Sparse-View 3D Gaussian Splatting"
_NeurIPS.cc/2025/Conference — NeurIPS 2025 poster_

### Official Review · Reviewer_jEws · 2025-06-05

**Clarity:** 3
**Significance:** 3
**Originality:** 2
**Rating:** 6
**Confidence:** 4

**Summary:**

This paper focuses on sparse view reconstruction. To be specific, they investigate the appearance artifacts in sparse-view 3DGS. These artifacts is generated due to the nature of volume rendering in 3DGS. The authors propose a Co-Adaptation Score to measure the difference between real appearance distribution and renderings. The authors propose two strategies to mitigate the co-adaptation in sparse-view 3DGS: (1) Gaussian dropout; (2) noise injection for opacity. The experiments show the effectiveness of the proposed method.

**Questions:**

In Figure 3, to the images in Var(), why are they all the same?
In Table 1, why are there no results of Binocular3DGS on DTU?

**Ethical Concerns:**

["NO or VERY MINOR ethics concerns only"]

**Final Justification:**

The rebuttal is clear and successfully addresses my concerns.

**Limitations:**

The novelty is quite limited. Also, the proposed method can be more clearly explained.
I am very interested in whether multi-view learning can improve sparse view reconstruction or not.
I anticipate an innovative 3D Gaussian filtering technique for sparse view reconstruction.
Although our current score is low, I would like to raise my score if authors address my concerns.

**Quality:**

3

**Strengths And Weaknesses:**

# Strengths
1. This paper is well organized and easy to follow.

2. This paper proposes a co-adaptation concept for sparse view reconstruction, which aims to quantify the difference between the real appearance distribution and Gaussian renderings.

3. The proposed metric well reflects the drawbacks of existing methods.

# Weakness

1. In Figure 4, the column c can be divided into two parts to make it clearer.

2. In Figure 5, the legends are too small to read.

3. The description of Opacity Noise Injection is unclear. For example, in Eq. 6, how to define $\sigma$? Moreover, the authors have introduced that different noise acting on different 3D Gaussian attributes have different impacts. Why do authors only inject noise into opacity? In theory, I think the noise should be very small, otherwise, it will degrade rendering results. Furthermore, the final opacity should be in the range [0, 1]. These details are not mentioned.

4. The proposed Dropout Regularization is confusing to me. Why randomly dropout but not find unreasonable 3D Gaussians and drop them? In theory, a tailored filtering method would be more effective and interesting, not randomly removing.

5. Overall, the novelty of this paper is quite limited. Although I recognize the findings of the co-adaptation concept, the proposed two regularization methods are too simple and easy.

6. To the sparse view reconstruction, multi-view learning [1] may be a good way to mitigate the co-adaptation problem.

[1] https://arxiv.org/abs/2410.02103

---

> ### Author Rebuttal · Authors · 2025-07-30
>
> We sincerely thank you for the thoughtful and constructive feedback. We're encouraged by your recognition of our co-adaptation concept and metric, and truly appreciate your openness to raising the score upon clarification. Below we address each concern in detail.
>
> ---
>
> **W1: In Figure 4, the column c can be divided into two parts to make it clearer.**
>
> **A1:** We appreciate the suggestion. In the revised version, we will clearly separate the DNGaussian and Binocular3DGS results in subfigure (c) of Figure 4 to improve readability and visual clarity.
>
> **W2: In Figure 5, the legends are too small to read.**
>
> **A2:** We agree the legends in Figure 5 are too small due to layout compression. We will enlarge them and adjust the layout in the revised version for better readability.
>
> **W3: The description of Opacity Noise Injection is unclear ...**
>
> **A3:** Thank you for these detailed questions. Our responses are as follows:
>
> (1) How is the noise scale parameter $\sigma$ in Eq. 6 defined?
>
> We set $\sigma$ to 0.8 in our experiments. Related ablation results can be found in Table 5.
>
> (2) Why only inject noise into opacity?
>
> We tested noise injection on multiple Gaussian attributes:
>
> - **Opacity:** Multiplicative noise effectively reduces co-adaptation and improves novel view quality without degradation.
> - **Scale:** Helps generalization but may cause blurring with excessive noise.
> - **Position (xyz):** Additive noise, scaled by local nearest-neighbor distance, often causes training divergence due to geometric instability.
> - **SH coefficients:** No significant improvement observed.
>
> We highlight opacity noise due to its best performance. More results are in Supplementary Section C.1.
>
> (3) Should the noise be small?
>
> Yes for position; even small noise is harmful. Opacity is more tolerant—$\sigma = 0.8$ works well (Table 5), despite being relatively large.
>
> (4) Should final opacity be clamped to [0, 1]?
>
> Yes. Opacity is clamped to [0, 1] after noise injection. We will clarify this in the revised version.
>
> **W4: .. Why randomly dropout but not find unreasonable 3D Gaussians and drop them? ..**
>
> **A4:** This is an excellent suggestion. We would like to clarify first that the goal of our dropout regularization is not to prune Gaussians, but to regularize the model to alleviate co-adaptation, particularly under sparse-view settings. Nevertheless, we have also explored more targeted dropout strategies beyond random dropout, including:
>
> (1) **Concrete Dropout [1]**
>
> In our implementation, each 3D Gaussian is assigned a learnable dropout probability $p_i$ in the range (0, 1). During training, we adopt the Binary Concrete distribution [1] to make the dropout process differentiable. Specifically, for each Gaussian, we sample a uniform variable $u_i$ from $Uniform(0, 1)$ and compute a soft mask $z_i$ as:
>
> $z_i = Sigmoid\left(
> \frac{
> \log(p_i) - \log(1 - p_i) + \log(u_i) - \log(1 - u_i)
> }{
> \tau
> }
> \right)$
>
> where $\tau$ is the temperature parameter (we use $\tau = 0.1$). This soft mask controls the probability of dropping each Gaussian. The final opacity $o_i$ is updated as:
>
> $o_i \leftarrow o_i \cdot (1 - z_i)$
>
> This formulation enables end-to-end training while softly dropping out Gaussians based on learned uncertainty. Additionally, we add a weak regularization term to the loss to avoid $p_i$ collapsing to extreme values (close to 0 or 1), which improves stability and preserves the intended randomness. However, concrete dropout did not yield the expected gains. We suspect that the soft mask lacks sufficient regularization strength under sparse data, leading to unstable learning and weak suppression of co-adaptation. See results in the table below.
>
>
> (2) **Density-based Dropout**
>
> We adapt dropout probabilities based on Gaussian point density. Specifically, we compute the nearest-neighbor distance for each Gaussian and rank all Gaussians accordingly. Points in denser regions are assigned higher dropout probabilities, as over co-adaptation is more likely in such areas. In contrast, sparse regions are less prone to overfitting and thus have lower dropout rates. In practice, we set dropout probabilities to range from 0.2 (sparsest) to 0.5 (densest) via linear mapping.
>
> (3) **Geometry-aware Dropout**
>
> We hypothesize that Gaussians in geometrically complex regions are more vulnerable to over co-adaptation and thus require stronger regularization. To approximate local complexity, we evaluated structural variation across each Gaussian’s nearest neighbors. However, this approach involves computing high-order geometric relationships across multiple neighbors for every point, resulting in prohibitive computational cost—approximately $\mathcal{O}(N^3)$ for $N$ Gaussians. Due to this inefficiency, we ultimately did not include this variant in our ablation studies.
>
> ### Comparison of Various Dropout Strategies
>
> We present a comparative study of the dropout strategies described above, based on the LLFF dataset and the Binocular3DGS baseline.
>
> | Setting   | PSNR ⬆️ | SSIM ⬆️ | LPIPS ⬇️ | Train CA ⬇️ | Test CA ⬇️ | Final GS Count (Avg.) |
> |-|-|-|-|-|-|-|
> | baseline  | 21.44   | 0.751   | 0.168 | 0.001845  | 0.001951    | 102021 |
> | w/ random dropout   | 22.12   | 0.777   | 0.154    | 0.000875     | 0.000978    | 122340  |
> | w/ concrete dropout    | 20.876  | 0.731   | 0.205    | 0.004680     | 0.005320    | 22959   |
> | w/ density-based dropout | 22.05   | 0.773   | 0.158    | 0.000478     | 0.000536    | 150968  |
> | w/ geometry-aware dropout | –  | –  | –  | –  | – | –  |
>
> >*Note for geometry-aware dropout. Too costly: computational complexity estimated as $\mathcal{O}(N^3)$.*
>
> From the results above, we observe that random dropout achieves the best overall performance across metrics. Nevertheless, more targeted dropout strategies remain a promising direction for future research. Our exploration of more targeted dropout strategies (e.g., gradient-based) will be included in the revised version of the paper.
>
>
> **W5: Overall, the novelty of this paper is quite limited ...**
>
> **A5:** We appreciate the reviewer’s acknowledgment of the co-adaptation finding. While the proposed regularization strategies—random Gaussian dropout and opacity noise injection—are intentionally simple, their novelty lies not in algorithmic complexity but in their ability to uncover and alleviate a previously unformalized problem in 3DGS: over co-adaptation under sparse-view settings. To our knowledge, this phenomenon has not been explicitly studied in the 3DGS literature. Our work offers a new lens to examine and improve generalization in sparse-view 3D reconstruction, and we believe this conceptual contribution can spark further research in more adaptive view-independent 3D representations, better initialization or densification schemes, or point-level semantic reasoning with splats. Despite their simplicity, the methods yield measurable improvements (both quantitative and qualitative) with negligible overhead. We hope this balance between effectiveness and simplicity may benefit the community in both research and practice.
>
>
> **W6: To the sparse view reconstruction, multi-view learning (MVGS) may be a good way to mitigate the co-adaptation problem.**
>
> **A6:** We thank the reviewer for the insightful suggestion regarding multi-view learning (MVGS). We implemented the official MVGS under sparse-view settings (3 views) and observed a significant performance drop—likely because MVGS was designed for dense-view reconstruction and lacks sparse-view adaptation.
>
> To further assess this, we added a same multi-view learning module atop vanilla 3DGS, training with three synchronized views per iteration. This setup accelerates convergence—likely due to stronger gradients—but lacks regularization, leading to overfitting, degraded performance, and a rapid increase in Gaussian count due to aggressive densification.
>
> Supplementary sparse-view MVGS results are shown below:
>
> | Method | PSNR ⬆️ | SSIM ⬆️ | LPIPS ⬇️ |
> |-|-|-|-|
> | MVGS  | 15.45 | 0.507 | 0.362 |
>
> To further evaluate the integration of multi-view learning (MV=3) into sparse-view 3DGS, we present results at different training iterations (1k, 3k, 5k, 10k):
>
> | Setting  | PSNR ⬆️ | SSIM ⬆️ | LPIPS ⬇️ | Train CA ⬇️ | Test CA ⬇️ | GS Count (Avg.) |
> |-|-|-|-|-|-|-|
> | 3DGS (10k)| 19.36   | 0.651   | 0.232 | 0.007543     | 0.008206 | 111137 |
> | w/ MV (10k) | 18.75 | 0.639| 0.245| 0.016841|0.024286 | 1144360 |
> | w/ MV (5k)  | 18.97 | 0.646| 0.240| 0.018252|0.022790 | 635671 |
> | w/ MV (3k)  | 19.41| 0.665| 0.228| 0.011994|0.013679| 259014 |
> | w/ MV (1k)  | 19.55 | 0.656| 0.292| 0.008352| 0.008635 | 22910 |
>
> While multi-view supervision accelerates early convergence, it does not reduce CA scores or sparse-view artifacts. This suggests multi-view learning needs further regularization for sparse views, and the direction remains promising.
>
> **Q7: In Figure 3, to the images in Var(), why are they all the same? In Table 1, why are there no results of Binocular3DGS on DTU?**
>
> **A7:** We recommend the reviewer to zoom in and examine the rendered images in Figure 3 more closely. Although they may appear visually similar at a glance, each image is rendered from a different Gaussian subset and contains observable differences, especially in opacity distribution and fine structural details within the same regions. We will consider enlarging the images in the final version to make such differences more clearly visible.
>
> Binocular3DGS (w/ both) on DTU is omitted in Table 1 as we used LLFF to show both strategies are similarly effective. Below are DTU results under joint training.
>
> | Setting | PSNR ⬆️ | SSIM ⬆️ | LPIPS ⬇️ | Train CA ⬇️ | Test CA ⬇️ |
> |-|-|-|-|-|-|
> | w/ both | 21.05 | 0.875 | 0.109 | 0.000736 | 0.001143 |
>
> **Reference**
> [1] Gal, Yarin, Jiri Hron, and Alex Kendall. "Concrete dropout." Advances in neural information processing systems 30 (2017).
>
> ---
>
> We hope our reply addresses your concerns and welcome any further discussion~

---

> > ### Comment · Reviewer_jEws · 2025-08-03
> > **Good Rebuttal**
> >
> > Thank the authors for their rebuttal. This rebuttal strikes my mind. It addresses my concerns.
> > I still believe Multi-view training would be useful and helpful for sparse-view tasks. I suggest that authors cite it in the camera-ready version.
> > I would like to raise my score to **strong accept**.

---

> > > ### Author Response · Authors · 2025-08-03
> > > **Thank you sincerely for your time and thoughtful feedback.**
> > >
> > > Thank you sincerely for your time and thoughtful feedback.
> > >
> > > We appreciate your recognition of our rebuttal and your insightful suggestion. We agree that multi-view training is a meaningful direction, and believe it holds promise even under sparse-view settings for mitigating novel view artifacts. In future work, we plan to explore more targeted regularization strategies that can be combined with multi-view training to further reduce co-adaptation in sparse-view reconstruction. We will include a citation to MVGS and a brief discussion in our revised version.
> > >
> > > Thank you again for your kind support.

---

### Official Review · Reviewer_AmxR · 2025-06-23

**Clarity:** 4
**Significance:** 4
**Originality:** 4
**Rating:** 5
**Confidence:** 5

**Summary:**

The paper presents a systematic analysis and improvement of appearance artifacts in 3D Gaussian Splatting for novel view synthesis under sparse-view settings. The authors find that existing 3DGS methods are prone to co-adaptation among Gaussian points during sparse-view training, which leads to abnormal artifacts when rendering from novel viewpoints. To address this, the authors introduce the Co-Adaptation Score (CA) as a metric to quantify the dependencies between Gaussians, and proposes two plug-and-play regularization strategies—Random Gaussian point dropout and opacity noise injection—to effectively suppress co-adaptation. These strategies enhance the model’s generalization and rendering quality for unseen views. Experimental results demonstrate that these methods significantly reduce artifacts and improve synthesis performance.

**Questions:**

See in the weakness part.

**Ethical Concerns:**

["NO or VERY MINOR ethics concerns only"]

**Final Justification:**

The author's response has addressed my concerns, so I have decided to accept the paper.

**Limitations:**

yes

**Paper Formatting Concerns:**

not notied

**Quality:**

4

**Strengths And Weaknesses:**

Strengths:
The authors systematically analyze the appearance artifact problem in novel view synthesis of sparse-view 3D Gaussian Splatting, clearly identifying the co-adaptation among Gaussians as a performance bottleneck. They propose a quantitative metric, the Co-Adaptation Score (CA), providing a tool for quantifying and analyzing this issue. The methods are simple and effective, introducing two plug-and-play regularization strategies—Random Gaussian point dropout and Opacity noise injection. The proposed methods are easy to integrate into existing 3DGS frameworks and can effectively suppress co-adaptation, thereby improving the model's generalization ability and rendering quality for novel views.

Weaknesses:
- The paper mainly focuses on its own proposed strategies, with limited horizontal comparison and discussion with other existing methods for mitigating appearance artifacts or improving generalization.
- It is unclear whether the proposed methods are effective for geometry reconstruction based on GS under sparse-view settings.

---

> ### Author Rebuttal · Authors · 2025-07-30
>
> We sincerely thank you for taking the time to read our paper and for your valuable comments. We truly appreciate your positive assessment of our work. Below, we address the two weaknesses you mentioned.
>
> ---
>
> **W1: The paper mainly focuses on its own proposed strategies, with limited horizontal comparison and discussion with other existing methods for mitigating appearance artifacts or improving generalization.**
>
> **A1:** Thank you for pointing out this limitation. In addition to our two proposed strategies we conducted further experiments with **three additional strategies** to better situate our method within the broader context. All results are based on the same baseline implementation (Binocular3DGS [1]) on LLFF:
>
> - **baseline + scale noise**: We inject multiplicative noise into the scale parameters of Gaussians during training. This scale noise also helps mitigate co-adaptation in sparse-view 3D reconstruction to some extent, leading to improved rendering quality.
> - **baseline + AbsGS [2]**: Applies pixel-wise absolute gradient accumulation for densification instead of standard gradient in vanilla 3DGS, which enhances fine details and reduces artifacts. The introduction of AbsGS also helps alleviate co-adaptation to some extent and improves rendering quality.
> - **baseline w/o opacity decay [1]**: We remove the default opacity decay strategy (a per-iter multiplicative factor 0.995 applied to all Gaussian opacities). Without this regularization, the co-adaptation score increases significantly, causing visible floater artifacts and worse rendering quality.
>
> | Setting                    | PSNR ⬆️ | SSIM ⬆️ | LPIPS ⬇️ | Train CA ⬇️ | Test CA ⬇️ |
> |---------------------------|--------|--------|---------|------------|-----------|
> | baseline [1]                 | 21.44  | 0.751  | 0.168   | 0.001845   | 0.001951  |
> | w/ dropout                | 22.12  | 0.777  | 0.154   | 0.000875   | 0.000978  |
> | w/ opacity noise          | 22.12  | 0.780  | 0.155   | 0.000660   | 0.000762  |
> | w/ scale noise    | 21.68 | 0.766  | 0.159   | 0.000927   | 0.001014  |
> | w/ AbsGS [2]      | 21.61  | 0.754  | 0.163   | 0.001446   | 0.001593  |
> | w/o opacity decay [1]| 20.26  | 0.708  | 0.202   | 0.004413   | 0.004830  |
>
> **W2: It is unclear whether the proposed methods are effective for geometry reconstruction based on GS under sparse-view settings.**
>
> **A2:** Thank you for pointing out this important aspect. To evaluate how our proposed strategies affect geometry reconstruction under sparse views, we compare the depth maps predicted by trained 3DGS models against pseudo ground-truth depth under an 18-view setting on LLFF.
>
> We use four common metrics:
>
> - **AbsRel**: Measures the average relative error between the predicted and ground-truth depths.
> - **RMSE**: Reflects the overall deviation between depth predictions and ground truth.
> - **MAE**: The average of absolute differences between depth predictions and ground truth.
> - **Log10**: Measures logarithmic error, which better captures errors in farther regions.
>
> The results are summarized below:
>
> | Setting            | AbsRel ⬇️ | RMSE ⬇️ | MAE ⬇️ | Log10 ⬇️ |
> |--------------------|-----------|--------|--------|----------|
> | baseline [3]           | 0.0732    | 4.2785   | 2.5802   | 0.0314   |
> | w/ dropout         | 0.0676    | 4.1048   | 2.4258   | 0.0295   |
> | w/ opacity noise   | 0.0651    | 3.9182   | 2.2923   | 0.0276   |
>
> As shown, both strategies consistently improve geometric accuracy across all metrics. In addition to these quantitative results, Supplementary Figure 1 provides visual comparisons of the reconstructed 3D point clouds. We observe significantly clearer structures, reduced color noise, and better point coherence.
>
> We appreciate the reviewer’s suggestion and will include more depth and geometry quality comparisons in the revised paper.
>
>
>
> **Reference**
> [1] Han, Liang, et al. "Binocular-guided 3d gaussian splatting with view consistency for sparse view synthesis." Advances in Neural Information Processing Systems 37 (2024): 68595-68621.
> [2] Ye, Zongxin, et al. "AbsGS: Recovering fine details in 3D Gaussian Splatting." Proceedings of the 32nd ACM International Conference on Multimedia. 2024.
> [3] Li, Jiahe, et al. "Dngaussian: Optimizing sparse-view 3d gaussian radiance fields with global-local depth normalization." Proceedings of the IEEE/CVF conference on computer vision and pattern recognition. 2024.
>
> ---
>
> We sincerely thank you again for your positive evaluation and recognition of our work. We hope the above responses have resolved your concerns. If you have any further questions or suggestions, we are more than happy to continue the discussion.

---

> > ### Comment · Reviewer_AmxR · 2025-08-02
> >
> > I thank the authors for their detailed response. I acknowledge their efforts and accept their answers. My recommendation remains the same.

---

> > > ### Author Response · Authors · 2025-08-03
> > > **Thank you very much for your time, recognition, and kind acknowledgment of our responses.**
> > >
> > > Thank you very much for your time, recognition, and kind acknowledgment of our responses. We truly appreciate the effort you put into carefully reading our paper and providing constructive feedback. Your comments and suggestions were encouraging and helpful for improving our work.

---

### Official Review · Reviewer_jvPz · 2025-07-01

**Clarity:** 3
**Significance:** 3
**Originality:** 3
**Rating:** 4
**Confidence:** 4

**Summary:**

This paper studies how different Gaussians co-adapt during training which can cause appearance artifacts when rendering from views not seen during training. The authors thoroughly analyze the phenomenon, which is especially prevalent in the few-view setup.
A metric, the co-adaptation (CA) score is proposed to measure this dependency, and it is defined by rendering views multiple times while randomly dropping some of the Gaussians and computing the variance. Two different methods are proposed to lower the CA score during training, and this is observed to also improve the rendering metrics of the test views.

**Questions:**

- A simple way to game the metric (i.e. achieving low training CA) is to simply duplicate each Gaussian N times with opacities scaled such that the total transmittance through all N duplicates is the same as the transmittance through the original one. For a large enough N this will result in a low variance when dropping each of the N gaussians with some probability, and hence a low CA. Was any behaviour like this observed? Specifically, what was the average number of Gaussians for the different datasets and methods (i.e. the rows in Table 1) after training with and without the proposed extra components (dropout and opacity noise)?
- Were any alternative measures tried that just measured variance in colors $c_i$, attempting to decouple it from however the opacities change? A straight-forward way would be to measure the variance of $c_i$ per pixel weighted proportionally to its rendering weight $\alpha_i \prod_{j=1}^{i-1}(1 - \alpha_j)$. How does this variance change with dropout or opacity noise compared to the baselines?
- Another way of thinking about how Gaussians should represent a scene is that rather than striving to have multiple redundant low opacity blobs (as in this paper - low CA is correlated with small alpha’s and that it should be robust to dropping random Gaussians) is to consider Gaussians as approximating e.g. a mesh with one or very few high opacity non-redundant Gaussians on the surfaces. Though in this way, a low CA is not desirable, but rather an overparametrization. Maybe a better way is to enforce a few high density Gaussians on surfaces, and then a large alpha will suppress colors that are off. Do you have any opinion about if/why the proposed CA-score is a better way of thinking than this?

**Ethical Concerns:**

["NO or VERY MINOR ethics concerns only"]

**Final Justification:**

The authors provided detailed answers to all my questions. The extra experiments regarding numbers of Gaussians and color variance strengthen the paper, since it shows that the lowered CA score depends mostly on reduced variance in colors and that the number of Gaussians per scene is roughly the same when training with dropout. I keep the borderline accept rating.

**Limitations:**

yes

**Quality:**

3

**Strengths And Weaknesses:**

Strengths:
- The authors analyse the reason for appearance artifact for novel view synthesis with Gaussian splatting, especially prevalent in the sparse view setup, and suggest that the issue is that multiple Gaussians co-adapt during training. The authors propose a metric by rendering multiple times while applying dropout to the Gaussians.
- The most important result is that the proposed methods to reduce CA score, in addition to indeed reducing the CA score, furthermore consistently improve the renderings of the test views.
The analysis of the CA score is mostly thorough and it is analysed in multiple aspects, e.g. the training dynamics and dependence on e.g. number of views or properties such as the degree of the spherical harmonics. It is also tested for multiple (5) baseline methods for Gaussian splatting.

Weaknesses:
- There are some limitations of the proposed metric. As shown in the supplementary, a low score is connected to having many Gaussians with low opacities as well as low variance in color ($CA \approx p(1-p) \sum_i (c_i \alpha_i)^2$, eq. (8) in the supplementary), and this is insufficiently investigated in the paper. Specifically, the coupling between colors and opacities makes it unclear if a low value is due to low variance in color or many low opacity Gaussians. Please see the items under the questions section.
- There are some straight-forward visualisations that are missing. For instance, plotting the CA score per pixel along with the renderings in e.g. fig. 6 or 7 would be informative to see for which pixels or regions there is low or high co-adaptation. Another less direct option would be e.g. the average or maximum alphas per pixel for the closest Gaussians, since the CA score is closely related to the average $\alpha_i c_i$ as shown in the supplementary.

---

> ### Author Rebuttal · Authors · 2025-07-30
>
> We sincerely thank you for the thoughtful and in-depth feedback. Your insightful questions and constructive suggestions—particularly regarding the limitations and alternative formulations of the CA score—have been invaluable in helping us further refine both the theoretical and practical aspects of our work. Below, we address each of your concerns in detail and discuss the directions your comments have inspired.
>
> ---
>
> **W1 and Q1:** The reviewer is concerned that the reduction in CA-score (Co-Adaptation score) observed after applying the proposed dropout and opacity noise strategies might be an artifact. Specifically, they suspect that our method could be “gaming” the CA metric by duplicating each Gaussian into multiple low-opacity ones, which artificially reduces color variance under dropout. They request evidence that such duplication behavior is not occurring, and ask for the average number of Gaussians across all settings.
>
>
> **Response:** We sincerely thank the reviewer for this insightful concern. To directly address this, we report the **final average number of Gaussians** trained under all settings (vanilla 3DGS, Binocular3DGS, and our proposed strategies). As shown in the table below, the number of Gaussians remains largely consistent across methods, and in some cases (e.g., 3DGS + opacity noise), the count is even slightly lower:
>
> | Setting             | PSNR ⬆️  | SSIM ⬆️ | LPIPS ⬇️ | Train CA ⬇️ | Test CA ⬇️ | Final GS Count (Avg.) |
> |---------------------|--------|--------|--------|------------|-----------|------------------------|
> | 3DGS                | 19.36  | 0.651  | 0.232  | 0.007543   | 0.008206  | 111137                 |
> | w/ dropout          | 20.20  | 0.691  | 0.211  | 0.001752   | 0.002340  | 115046                 |
> | w/ opacity noise    | 19.91  | 0.676  | 0.223  | 0.001531   | 0.002300  | 98356                  |
> | Binocular3DGS       | 21.44  | 0.751  | 0.168  | 0.001845   | 0.001951  | 102021                 |
> | w/ dropout          | 22.12  | 0.777  | 0.154  | 0.000875   | 0.000978  | 122340                 |
> | w/ opacity noise    | 22.12  | 0.780  | 0.155  | 0.000660   | 0.000762  | 104832                 |
>
> These results indicate that our method does not reduce the CA-score by duplicating Gaussians, as the final point count remains comparable or even lower. This supports the validity of CA as a meaningful metric and reflects genuine improvements in opacity and color distribution. We will include these results in the revised version to further clarify this point.
>
> ---
>
> **W2:** Missing co-adaptation score map visualizations. The reviewer points out that our paper lacks visualizations of co-adaptation (CA) scores, and suggests providing per-pixel CA heatmaps or related indicators such as alpha distributions to better illustrate regions of high or low co-adaptation.
>
> **Response:** We thank the reviewer for the valuable suggestion. We agree that CA visualizations can offer intuitive insight into how co-adaptation varies spatially. Preliminary visualizations indicate that regions with high CA scores often align with geometrically complex areas and tend to have more densely packed or overlapping Gaussians. In the revised version, we will include per-pixel CA score heatmaps overlaid with rendered views (e.g., Figure 6 or 7).
>
> ---
>
> **Q2:** Were any alternative measures tried that just measured variance in colors $c_i$, attempting to decouple it from however the opacities change? A straightforward way would be to measure the variance of $c_i$ per pixel weighted proportionally to its rendering weight $w_i = \alpha_i \cdot \prod_{j=1}^{i-1}(1 - \alpha_j)$. How does this variance change with dropout or opacity noise compared to the baselines?
>
> **Response:** Thank you for the insightful suggestion. We have implemented the proposed alternative variance measure in our CUDA rendering kernel by computing, for each pixel, the **weighted variance** of contributing color values $c_i$, using their compositing weights:
>
> $$
> w_i = \alpha_i \cdot \prod_{j=1}^{i-1}(1 - \alpha_j)
> $$
>
> Specifically, we compute the per-pixel variance of the color parameters ${c_i}$, which can be derived as follows:
>
> $$
> \mathrm{Var}(c) = \frac{\sum_i w_i c_i^2}{\sum_i w_i} - \left( \frac{\sum_i w_i c_i}{\sum_i w_i} \right)^2
> $$
>
> This formulation reflects the variance of color contributions decoupled from however the opacities change. We compute the **color variance metric (CV)** by averaging the per-pixel variance of the color parameters ${c_i}$ over the entire rendered image.
>
> The results are summarized below:
>
> | Setting           | PSNR ⬆️ | Train CA ⬇️ | Test CA ⬇️ | Train CV ⬇️ | Test CV ⬇️ | Final GS Count (Avg.)  |  GS Radius (Avg.)  |
> |------------------|----------|---------------|--------------|------------------------|-----------------------|---|---|
> | baseline          | 21.44     | 0.001845      | 0.001951     | 0.2496                 | 0.2563                | 102021 | 21.38 |
> | w/ dropout        | 22.12     | 0.000875      | 0.000978     | 0.0992                 | 0.1065                | 122340 | 35.93 |
> | w/ opacity noise  | 22.12    | 0.000660      | 0.000762     | 0.04760                | 0.0534                | 104832 | 31.64 |
>
> We observe from the above results that both dropout and opacity noise significantly reduce the **color variance metric (CV)** and **co-adaptation score (CA)**. The CV reduction from dropout is weaker than that from opacity noise, possibly because dropout tends to produce slightly more Gaussians with larger scales after training. This leads to a higher color variance.
>
> We would also like to share a related experiment we previously explored: instead of computing the CV metric, we once tried adding a weak constraint during training by aligning the color directions (after normalization) of all Gaussians with GT pixel color along the same ray. However, even with very small loss weights, this significantly degraded the rendering quality. We believe this is because such constraints overly limit the expressive power of 3DGS, where color co-adaptation is part of its inherent flexibility.
>
> Thank you for raising this insightful point. We will incorporate the implementation details of the CV metric, the associated experimental findings, and our related exploratory attempts into the revised version of the manuscript.
>
> ---
>
> **Q3:** **On whether enforcing fewer high-opacity Gaussians is a better representation approach than minimizing CA-score.**
>
> **Response:** Thank you for this insightful perspective. We agree that, in theory, having a few high-opacity Gaussians located precisely on surfaces is an ideal and compact representation. However, under sparse-view settings, achieving such configurations is extremely difficult. In practice, we observe that surface Gaussians tend to be more concentrated and have relatively higher opacities, but not necessarily fewer in number.
>
> More importantly, unlike traditional mesh or point cloud representations, co-adaptation is an inherent property of 3DGS. That is, multiple Gaussians collaboratively contribute to the color fitting along a ray, which reflects a key feature of the 3DGS representation. In this context, our proposed Co-Adaptation (CA) score offers a principled way to quantify how strongly Gaussians rely on each other in the rendering process. We believe that CA provides a more faithful and flexible characterization of this intrinsic cooperative behavior, rather than imposing strict sparsity or singular high-opacity constraints, which may undermine the expressive power of the 3DGS framework.
>
> That said, we also acknowledge the potential value in exploring more **compact and cohesive representations** that retain the high expressiveness of 3DGS while naturally reducing unnecessary co-adaptation. Developing such representations—ones that maintain structural efficiency without over-constraining the cooperative dynamics—could be a promising direction for future research.
>
> ---
>
> We hope our response has addressed your concerns accurately and respectfully, and that we have not misunderstood the valuable perspective you intended to convey. If you have further thoughts or suggestions, we would be truly happy to continue the discussion. Thank you again for your thoughtful and inspiring feedback.

---

> > ### Comment · Reviewer_jvPz · 2025-08-06
> >
> > The authors provided detailed answers to all my questions. The extra experiments regarding numbers of Gaussians and color variance strengthen the paper, since it shows that the lowered CA score depends mostly on reduced variance in colors and that the number of Gaussians per scene is roughly the same when training with dropout. I keep the accept rating.

---

> > > ### Author Response · Authors · 2025-08-07
> > > **Thank you very much for your time and effort in carefully reading our paper and providing constructive feedback.**
> > >
> > > Thank you very much for your time and effort in carefully reading our paper and providing constructive feedback. We truly appreciate your decision to maintain the acceptance score.
> > >
> > > In the final version, we will include:
> > > - The formula and results for the color variance metric;
> > > - Additional experiments on Gaussian counts;
> > > - Visualizations of the CA metric.
> > >
> > > These updates will help improve the clarity and completeness of our work.
> > > Thanks again for your insightful and supportive review.

---

### Official Review · Reviewer_S997 · 2025-07-03

**Clarity:** 3
**Significance:** 2
**Originality:** 2
**Rating:** 3
**Confidence:** 4

**Summary:**

his paper points out that the appearance artifacts in sparse-view 3DGS from excessive entanglement (co-adaptation) among Gaussians. That is, Gaussians form strong dependent relationships to fit the training views, ignoring the real appearance distribution of the scene. Therefore, Co-Adaptation Score (CA) is proposed to quantify the degree of entanglement among Gaussians by calculating the pixel-wise variance of rendering results from different Gaussian subsets under the same viewpoint. Experiments show that the CA score naturally decreases as the number of training views increases. The author also proposed two strategies, random gaussian dropout and opacity noise injection, to mitigate the problem.

**Questions:**

Please refer to the Strengths And Weaknesses section.

**Ethical Concerns:**

["NO or VERY MINOR ethics concerns only"]

**Limitations:**

Yes.

**Paper Formatting Concerns:**

None.

**Quality:**

3

**Strengths And Weaknesses:**

Strengths:

1. This paper establishes a connection between the sparse-view artifacts observed in 3D Gaussian Splatting (3DGS) and the co-adaptation effect. The proposed Co-Adaptation (CA) metric offers a quantifiable tool for analyzing this phenomenon.

2. The paper introduces two simple yet versatile mitigation strategies that are compatible with various 3DGS variants. Both quantitative metrics (PSNR, SSIM, LPIPS) and qualitative results demonstrate that these strategies effectively reduce CA scores and alleviate artifacts.

Weaknesses:
1. The core idea draws inspiration from dropout, which is conceptually straightforward. As a result, the level of novelty appears limited.

2. Although the paper introduces a new method aimed at improving the generalization ability of 3DGS under sparse-view settings, the overall improvement is relatively modest.

3. The authors state that combining both proposed strategies does not lead to further improvement, but they do not provide experimental evidence to support this claim.

4. Since the dropout strategy improves generalization by reducing the number of Gaussians during rendering, it would be helpful to compare this approach with directly reducing the number of points at initialization to verify whether similar benefits can be achieved.

5. While the focus is on sparse-view training, the paper lacks evaluation under extremely limited-view settings (e.g., 1–2 input views), which would help better assess the robustness of the proposed strategies under severe data scarcity.

---

> ### Author Rebuttal · Authors · 2025-07-29
>
> We sincerely thank you for the time and effort dedicated to reading our paper and providing constructive feedback. We have carefully considered your concerns and respond to them point by point below.
>
> ---
>
> **W1: The core idea draws inspiration from dropout, which is conceptually straightforward. As a result, the level of novelty appears limited.**
>
> **A1:** While dropout is a well-established concept in neural networks, its application in 3D representation learning remains underexplored. More importantly, our work does not simply reuse dropout, but leverages it to **reveal and mitigate a unique co-adaptation phenomenon in 3D Gaussian Splatting (3DGS)**—a problem that, to our knowledge, has not been previously formalized. We believe this insight may help the community better understand the distinctive behaviors of 3DGS, and potentially inspire further studies on point initialization, densification, per-Gaussian feature learning (e.g., for semantic tasks), and more generalizable 3D representations beyond specific view settings. In this sense, our contribution extends beyond the method itself and offers a new perspective for future exploration.
>
> **W2: Although the paper introduces a new method aimed at improving the generalization ability of 3DGS under sparse-view settings, the overall improvement is relatively modest.**
>
> **A2:** We acknowledge that our proposed methods—random Gaussian dropout and opacity noise injection—are simple by design. However, they are highly practical and lightweight, making them easy to plug into existing sparse-view 3DGS frameworks to improve novel view generalization with minimal overhead. Our focus is on **addressing appearance artifacts caused by over co-adaptation**, a pervasive issue in sparse-view settings that has been largely overlooked. Despite their simplicity, the improvements are not negligible: quantitative results demonstrate reliable gains across several 3DGS baselines, and more importantly, qualitative comparisons reveal visibly cleaner point cloud geometry and significantly reduced color noise. We kindly refer the reviewer to Supplementary Figure 1 for a clearer illustration of the benefits.
>
> **W3: The authors state that combining both proposed strategies does not lead to further improvement, but they do not provide experimental evidence to support this claim.**
>
> **A3:** We respectfully clarify that our paper includes quantitative and qualitative evidence supporting this claim. In Table 1 of the main paper (last row), we show results on the LLFF dataset using the Binocular3DGS baseline, where applying both strategies together achieves metrics similar to using either one individually. Additionally, in the DTU dataset, the results with both strategies (as shown below) remain comparable to single-strategy settings, reinforcing our observation:
>
> | Setting  | PSNR ⬆️ | SSIM ⬆️ | LPIPS ⬇️ | Train CA ⬇️ | Test CA ⬇️ |
> |----------|--------|--------|---------|-------------|------------|
> | w/ both  | 21.05  | 0.875  | 0.109   | 0.000736    | 0.001143   |
>
> This supports our point that while the two strategies are effective, they both address the same underlying co-adaptation issue, and thus do not provide additive improvements when combined. Furthermore, in nearly all qualitative comparisons, we separately present the results of applying either random Gaussian dropout or opacity noise injection. The visual improvements—such as clearer geometry and reduced artifacts—are highly similar across both cases, further suggesting that the two strategies address the same underlying co-adaptation issue.
>
> **W4: Since the dropout strategy improves generalization by reducing the number of Gaussians during rendering, it would be helpful to compare this approach with directly reducing the number of points at initialization to verify whether similar benefits can be achieved.**
>
> **A4:** To address this suggestion, we conducted a controlled comparison experiment by directly varying the number of initialization Gaussians in the DNGaussian framework. Specifically, we modified the `num_pts` parameter in the initialization code, which controls the total number of Gaussian points prior to training.
>
> As shown in the table below, we evaluate performance across different initialization sizes using the LLFF dataset. We find that simply reducing the number of initialization points (e.g., 10, 30, 50, etc.) does not replicate the generalization benefits brought by our random Gaussian dropout strategy. Although reducing initial points can slightly lower CA and the final Gaussian count, it leads to worse PSNR, SSIM, and LPIPS.
>
> In contrast, applying dropout during training (row *50 w/ dropout*) yields:
> - Highest PSNR (19.43) and SSIM (0.623)
> - Lowest Train/Test CA scores
> - A relatively compact final Gaussian count
>
> This demonstrates that **dropout offers more than static reduction**—it provides dynamic regularization that actively suppresses co-adaptation and improves generalization under sparse views.
>
> ### Initialization Comparison on LLFF (DNGaussian baseline)
>
> | **N**               | **PSNR ⬆️** | **SSIM ⬆️** | **LPIPS ⬇️** | **Train CA ⬇️** | **Test CA ⬇️** | **Final GS Count (Avg.)** |
> |---------------------|-------------|-------------|--------------|------------------|------------------|----------------------------|
> | 10                  | 18.37       | 0.571       | 0.331        | 0.007134         | 0.007515         | 39006                      |
> | 30                  | 18.68       | 0.588       | 0.307        | 0.007179         | 0.007695         | 43743                      |
> | 50                  | 18.93       | 0.599       | 0.295        | 0.007234         | 0.007645         | 45835                      |
> | **50 (w/ dropout)** | **19.43**   | **0.623**   | 0.302    | **0.003242**     | **0.003821**     | 42113                      |
> | 70                  | 18.85       | 0.591       | 0.297        | 0.007157         | 0.007646         | 48655                      |
> | 90                  | 18.75       | 0.592       | 0.295        | 0.007268         | 0.007703         | 51494                      |
> | 100                 | 18.84       | 0.594       | 0.293        | 0.007191         | 0.007721         | 52152                      |
> | 500                 | 18.72       | 0.584       | **0.291**    | 0.007232         | 0.007955         | 97329                      |
> | 1000                | 18.73       | 0.571       | 0.296        | 0.007488         | 0.008277         | 160211                     |
>
> > *Note: The `num_pts` is controlled via `num_pts = int(pcd_shape.max() * N)` in code; the row **50 (w/ dropout)** shares the same initial Gaussian count with N=50 but applies dropout during training.*
>
> **W5: While the focus is on sparse-view training, the paper lacks evaluation under extremely limited-view settings (e.g., 1–2 input views), which would help better assess the robustness of the proposed strategies under severe data scarcity.**
>
> **A5:** To address this concern, we conduct additional experiments on DNGaussian under extremely sparse settings with only 1–2 input views on the LLFF dataset. As shown in the table below, both of our proposed regularization strategies improve performance even under extremely limited-view conditions.
>
> | Views          | PSNR ⬆️ | SSIM ⬆️ | LPIPS ⬇️ | Train CA ⬇️ | Test CA ⬇️ |
> |----------------|--------|--------|---------|-------------|------------|
> | 1              | 12.30  | 0.287  | 0.520   | 0.009779    | 0.010383   |
> | w/ dropout     | 12.35  | **0.303**  | 0.529   | **0.004436**    | **0.005119**   |
> | w/ opacity noise | **12.45**  | 0.294  | **0.511**   | 0.005078    | 0.005712   |
> | 2              | 16.06  | 0.457  | 0.381   | 0.008123    | 0.008865   |
> | w/ dropout     | **16.78**  | **0.495**  | **0.379**   | 0.003641    | 0.004900   |
> | w/ opacity noise | 16.58  | 0.488  | 0.384   | **0.002952**    | **0.003880**   |
> | 3              | 18.93  | 0.599  | 0.295   | 0.007234    | 0.007645   |
> | w/ dropout     | **19.43**  | **0.623**  | 0.302   | **0.003242**    | **0.003821**   |
> | w/ opacity noise | 19.15  | 0.608  | **0.294**   | 0.004507    | 0.005071   |
>
> ---
>
> We hope the above response sufficiently addresses your concerns. If you have any further concerns or suggestions, we sincerely welcome your continued comments and discussion. Thank you again for your valuable feedback and time.

---

> ### Author Response · Authors · 2025-08-07
> **Dear Reviewer S997, we sincerely hope to receive your further thoughts or questions before the discussion deadline (Aug 8, 11.59pm AoE).**
>
> Dear Reviewer S997,
>
> Thank you again for your thoughtful review.
>
> As the discussion phase ends in two days, we are very anxious to hear back from you. Your comments raised several key concerns, and we’ve carefully addressed each of them with detailed responses and new experiments. However, we’re unsure if our replies resolved your doubts.
>
> Your feedback is extremely important to us — and your support would mean a great deal for the outcome of this submission. We sincerely hope to receive your further thoughts or questions before the discussion deadline.
>
> Warm regards,
> Authors of Submission 3276

---

### Note · Authors · 2025-08-12

**For Reviewer S997.** **Unfortunately, we have not received any follow-up from this reviewer during the discussion phase.** Regarding the raised concerns:
- **Novelty.** While the reviewer considers our approach simple, we believe *simple yet effective* and *plug-and-play* is an important merit. Our strategies can be directly applied to a variety of SOTA sparse-view 3DGS baselines, as evidenced by consistent improvements across benchmarks. Moreover, our definition, analysis, and quantification of the **co-adaptation phenomenon** in 3DGS offer **insight** that extends beyond the proposed strategies themselves. This perspective can inspire future research in areas such as **point-cloud initialization**, **Gaussian densification**, and **point-level feature learning**, potentially influencing a range of 3DGS-related tasks.
- **Quality gains.** We disagree that improvements are relatively modest. Supplementary point-cloud visualizations show **marked noise reduction** and **clearer structure** for both strategies.
- **No evidence for a claim.** The paper (Table 1 and main text) already reports the **combined** setting; the combined result is comparable to using either strategy alone.
- **On fewer initial Gaussians.** Our experiments show that merely reducing the initial point count **does not** reproduce the CA reduction achieved by our methods.
- **Extreme sparsity (1–2 views).** Per the request, we added results showing **improved novel-view quality** even under 1–2 input views.


**For reviewers jvPz, AmxR, and jEws,** we have received their feedback during the discussion phase, confirming that our rebuttal has fully addressed and clarified their concerns. Please refer to the rebuttal for detailed responses~

---

### Decision · Program_Chairs · 2025-09-17

**Decision:**

Accept (poster)

**Comment:**

The paper proposed to analyze the source of appearance artifacts and identify the entanglement among Gaussians as the performance bottleneck for sparse-view 3DGS. It further proposed a metric of Co-Adaptation Score to measure the entanglement among Gaussians in sparse-view 3DGS. It also proposed two plug-and-play training strategies that can suppress co-adaptation among gaussians.

The paper was reviewed by four reviewers in the community. The authors provided rebuttal and there had been discussions among the authors and the reviewers. The rebuttal and the following discussions succefully addressed most of the issues from the initial reviews where the reviewers acknowledged the improvement. Afterward, the recommendations from the reviewer were Borderline accept, Accept and Strong Accept. Reviewer S997 was with an initial Borderline Reject rating while the authors addressed most of the comments.

Given the consistent ratings, I would like to accept the paper. While the reviewers are consistent with the contributions from the paper, the authors are requested to revise the paper correspondingly.